# Adaptive First-Order Methods Revisited: Convex Optimization without Lipschitz Requirements

**Kimon Antonakopoulos**   **Panayotis Mertikopoulos**
Univ. Grenoble Alpes, CNRS, Inria, Grenoble INP, LIG 38000 Grenoble, France & Criteo AI Lab
kimon.antonakopoulos@inria.fr  panayotis.mertikopoulos@imag.fr

## Abstract

We propose a new family of adaptive first-order methods for a class of convex mini-mization problems that may fail to be Lipschitz continuous or smooth. Specifically, motivated by a recent flurry of activity on non-Lipschitz (NoLips) optimization, we consider problems that are continuous or smooth relative to a reference Bregman function – as opposed to a global, ambient norm (Euclidean or otherwise). These conditions encompass a wide range of problems with singular objectives that can-not be treated with standard first-order methods for Lipschitz continuous/smooth problems – such as Fisher markets, Poisson tomography problems, $D$-optimal design, and the like. In this setting, the application of existing order-optimal adap-tive methods – like UNIXGRAD or ACCELEGRAD – is not possible, especially in the presence of randomness and uncertainty. The proposed method – which we call *adaptive mirror descent* (ADAMIR) – aims to close this gap by concurrently achieving min-max optimal rates in problems that are relatively continuous or smooth, including stochastic ones.

## 1 Introduction

Owing to their wide applicability and flexibility, first-order methods continue to occupy the forefront of research in learning theory and continuous optimization. Their analysis typically revolves around two basic regularity conditions for the problem at hand: (*i*) Lipschitz continuity of the problem's objective, and / or (*ii*) Lipschitz continuity of its gradient (also referred to as *Lipschitz smoothness*). Depending on which of these conditions holds, the lower bounds for first-order methods with perfect gradient input are $\Theta(1/\sqrt{T})$ and $\Theta(1/T^2)$ after $T$ gradient queries, and they are achieved by gradient descent and Nesterov's fast gradient algorithm respectively [36, 37, 48]. By contrast, if the optimizer only has access to stochastic gradients (as is often the case in machine learning and distributed control), the corresponding lower bound is $\Theta(1/\sqrt{T})$ for both classes [14, 34, 37].

This disparity in convergence rates has led to a surge of interest in adaptive methods that can seamlessly interpolate between these different regimes. Two state-of-the-art methods of this type are the ACCELEGRAD and UNIXGRAD algorithms of Levy et al. [26] and Kavis et al. [24]: both algorithms concurrently achieve an $\mathcal{O}(1/\sqrt{T})$ value convergence rate in non-smooth problems, an $\mathcal{O}(1/T^2)$ rate in smooth problems, and an $\mathcal{O}(1/\sqrt{T})$ average rate if run with bounded, unbiased stochastic gradients (the smoothness does not affect the rate in this case). In this regard, UNIXGRAD and ACCELEGRAD both achieve a "best of all worlds" guarantee which makes them ideal as off-the-shelf solutions for applications where the problem class is not known in advance – e.g., as in online traffic routing, game theory, etc.

At the same time, there have been considerable efforts in the literature to account for problems that do not adhere to these Lipschitz regularity requirements – such as Fisher markets, quantum tomography, $D$-design, Poisson deconvolution / inverse problems, and many other examples [7, 10, 11, 28, 29, 45]. The reason that the Lipschitz framework fails in this case is that, even when the problem's domain

35th Conference on Neural Information Processing Systems (NeurIPS 2021).

is bounded, the objective function exhibits singularities at the boundary, so it cannot be Lipschitz continuous or smooth. As a result, no matter how small we pick the step-size of a standard gradient method (adaptive or otherwise), the existence of domains with singular gradients can – and typically *does* – lead to catastrophic oscillations (especially in the stochastic case).

A first breakthrough in this area was provided by Birnbaum et al. [10] and, independently, Bauschke et al. [7] and Lu et al. [29], who considered a "Lipschitz-like" gradient continuity condition for problems with singularities.[1] At around the same time, Lu [28] and Teboulle [45] introduced a "relative continuity" condition which plays the same role for Lipschitz continuity. Collectively, instead of using a global norm as a metric, these conditions employ a Bregman divergence as a measure of distance, and they replace gradient descent with *mirror descent* [14, 34].

In these extended problem classes, *non-adaptive* mirror descent methods achieve an $\mathcal{O}(1/\sqrt{T})$ value convergence rate in relatively continuous problems [2, 28], an $\mathcal{O}(1/T)$ rate in relatively smooth problems [7, 10], and an $\mathcal{O}(1/\sqrt{T})$ average rate if run with stochastic gradients in relatively continuous problems [2, 28]. Importantly, the $\mathcal{O}(1/T)$ rate for relatively smooth problems *does not match* the $\mathcal{O}(1/T^2)$ rate for standard Lipschitz smooth problems: in fact, even though [20] proposed a tentative path towards faster convergence in certain non-Lipschitz problems, Dragomir et al. [17] recently established an $\Omega(1/T)$ lower bound for problems that are relatively-but-not-Lipschitz smooth.

**Our contributions.** Our aim in this paper is to provide an *adaptive, parameter-agnostic* method that simultaneously achieves order-optimal rates in the above "non-Lipschitz" framework. By design, the proposed method – which we call *adaptive mirror descent* (ADAMIR) – has the following desirable properties:

1. When run with perfect gradients, the trajectory of queried points converges, and the method's rate of convergence in terms of function values interpolates between $\mathcal{O}(1/\sqrt{T})$ and $\mathcal{O}(1/T)$ for relatively continuous and relatively smooth problems respectively.

2. When run with stochastic gradients, the method attains an $\mathcal{O}(1/\sqrt{T})$ average rate of convergence.

3. The method applies to both constrained and unconstrained problems, without requiring a finite Bregman diameter or knowledge of a compact domain containing a solution.

The only thing we assume known in the above is the reference Bregman function with respect to which the problem's objective is relatively continuous/smooth; other than that, we assume no prior information on the problem's regularity class / modulus and/or the oracle model being accessed (deterministic or stochastic).[2] The enabling apparatus for these properties is an adaptive step-size policy in the spirit of [6, 18, 30]. However, a major difference – and technical difficulty – is that the relevant definitions cannot be stated in terms of global norms, because the variation of non-Lipschitz function explodes at the boundary of the problem's domain (put differently, gradients may be unbounded even over bounded domains). For this reason, our policy relies on the aggregation of a suitable sequence of "Bregman residuals" that stabilizes seamlessly when approaching a smooth solution, thus enabling the method to achieve faster convergence rates.

**Related work.** Beyond the references cited above, problems with singular objectives were treated in a recent series of papers in the context of online and stochastic optimization [2, 49]; however, the proposed methods are *not* adaptive, and *they do not interpolate* between different problem classes.

In the context of adaptive methods, the widely used ADAGRAD algorithm of Duchi et al. [18] and McMahan and Streeter [30] was recently shown to interpolate between an $\mathcal{O}(1/\sqrt{T})$ and $\mathcal{O}(1/T)$ rate of convergence [26, 27]. More precisely, Li and Orabona [27] showed that a specific, "one-lag-behind" variant of ADAGRAD with prior knowledge of the smoothness modulus achieves an $\mathcal{O}(1/T)$

---

[1]This condition was first examined by Birnbaum et al. [10] in the context of Fisher markets. The analysis of Bauschke et al. [7] and Lu et al. [29] is much more general, but several ideas are already present in [10].

[2]The reference function has to do with the type of gradient singularities that develop at the boundary of the problem's domain. For example, the Fisher market models that we discuss later in the paper always exhibit $\Theta(\log x)$ gradient singularities at the boundary, so the appropriate choice for such models is the negative entropy, and this is true whether the specific market model is everywhere differentiable or not. Likewise, Poisson inverse problems always exhibit $\Theta(1/x)$ gradient singularities at the boundary of the positive orthant, which means that the log-entropy is the appropriate reference function for such problems.

| | | Constr. / Uncon. | Stoch. (L) | RC | RS | Stoch. (R) |
|---|---|---|---|---|---|---|
| ADAGRAD | [18] | ✓/✓ | ✓ | ✗ | ✗ | ✗ |
| ACCELEGRAD | [26] | ✗/✓ | ✓ | ✗ | ✗ | ✗ |
| UNIXGRAD | [24] | ✓/✗ | ✓ | ✗ | ✗ | ✗ |
| UPGD | [39] | ✓/✓ | ✗ | ✗ | ✗ | ✗ |
| GMP | [44] | ✓/✓ | ✗ | ✗ | $1/T$ | ✗ |
| ADAPROX | [3] | ✓/✓ | ✗ | partial | partial | ✗ |
| ADAMIR | [ours] | ✓/✓ | ✓ | $1/\sqrt{T}$ | $1/T$ | $1/\sqrt{T}$ |

**Table 1:** Overview of related work. For the purposes of this table, (L) refers to "Lipschitz" and (R) to "relative" continuity or smoothness respectively. In the case of ADAPROX, "partial" means that the non-Lipschitz conditions under which it guarantees convergence form a subset of the RC/RS conditions. Logarithmic factors are ignored throughout; we also note that the $\mathcal{O}(1/T)$ rate in the RS column is, in general, unimprovable [17].

rate in smooth, unconstrained problems; concurrently, Levy et al. [26] obtained the same rate in a parameter-agnostic context. In either case, ADAGRAD achieves an $\mathcal{O}(1/\sqrt{T})$ rate of convergence in stochastic problems (though with somewhat different assumptions for the randomness).

In terms of rate optimality for Lipschitz smooth problems, ADAGRAD is outperformed by ACCELE-GRAD [26] and UNIXGRAD [24]: these methods both achieve an $\mathcal{O}(1/T^2)$ value convergence rate in Lipschitz smooth problems, and an $\mathcal{O}(1/\sqrt{T})$ rate in stochastic problems with bounded gradient noise. By employing an efficient line-search step, the *universal primal gradient descent* (UPGD) algorithm of Nesterov [39] achieves order-optimal guarantees in the wider class of Hölder continuous problems (which includes the Lipschitz continuous and smooth cases as extreme cases); however, UPGD does not cover stochastic problems or problems with relatively continuous / smooth objectives.

As far as we are aware, the closest works to our own are the *generalized mirror-prox* (GMP) algorithm of [44] and the ADAPROX method of [2], both designed for variational inequality problems. The GMP algorithm can achieve interpolation between different classes of Hölder continuous problems and can adapt to the problem's relative smoothness modulus, but it does not otherwise interpolate between the relatively smooth and relatively continuous classes. Moreover, GMP requires knowledge of a "domain of interest" containing a solution of the problem; in this regard, it is similar to ACCELEGRAD [26] (though it does not require an extra projection step). The recently proposed ADAPROX method of Antonakopoulos et al. [3] also achieves a similar interpolation result under a set of assumptions that are closely related – but not equivalent – to the relatively continuous/smooth setting of our paper. Moreover, neither of these papers covers the stochastic case; to the best of our knowledge, ADAMIR is the first method in the literature capable of adapting to relatively continuous / smooth objectives, even under uncertainty. For convenience, we detail these related works in Table 1 above.

## 2 Problem setup and preliminaries

**Problem statement.** Throughout the sequel, we will focus on convex minimization problems of the general form

$$
\begin{aligned}
\text{minimize} \quad & f(x), \\
\text{subject to} \quad & x \in \mathcal{X}.
\end{aligned}
\tag{Opt}
$$

In the above, $\mathcal{X}$ is a convex subset of a normed $d$-dimensional space $\mathcal{V} \cong \mathbb{R}^d$, and $f \colon \mathcal{V} \to \mathbb{R} \cup \{\infty\}$ is a proper lower semi-continuous (l.s.c.) convex function with $\operatorname{dom} f = \{x \in \mathcal{V} : f(x) < \infty\} = \mathcal{X}$. Compared to standard formulations, we stress that $\mathcal{X}$ *is not assumed to be compact, bounded, or even closed.* This lack of closedness will be an important feature for our analysis because we are interested in objectives that may develop singularities at the boundary of their domain; for a class of relevant examples of this type, see [1, 2, 7, 9–11, 22, 29, 49] and references therein.

To formalize our assumptions for (Opt), we will write $\partial f(x)$ for the *subdifferential* of $f$ at $x$, and $\mathcal{X}_\circ \equiv \operatorname{dom} \partial f = \{x \in \mathcal{X} : \partial f(x) \neq \varnothing\}$ for the *domain of subdifferentiability* of $f$. Formally, elements of $\partial f$ will be called subgradients, and we will treat them throughout as elements of the dual space $\mathcal{V}^*$ of $\mathcal{V}$. By standard results, we have $\operatorname{ri} \mathcal{X} \subseteq \mathcal{X}_\circ \subseteq \mathcal{X}$, and any solution $x^*$ of (Opt) belongs to $\mathcal{X}_\circ$ [42, Chap. 26]; to avoid trivialities, we will make the following blanket assumption.

**Assumption 1.** The solution set $\mathcal{X}^* \equiv \arg\min f \subseteq \mathcal{X}_\circ$ of (Opt) is nonempty.

Two further assumptions that are standard in the literature (but which we relax in the sequel) are:

1. *Lipschitz continuity:* there exists some $G > 0$ such that

$$|f(x') - f(x)| \leq G\|x' - x\| \quad \text{for all } x, x' \in \mathcal{X}. \tag{LC}$$

2. *Lipschitz smoothness:* there exists some $L > 0$ such that

$$f(x') \leq f(x) + \langle v, x' - x \rangle + \frac{L}{2}\|x' - x\|^2 \quad \text{for all } x, x' \in \mathcal{X} \text{ and all } v \in \partial f(x). \tag{LS}$$

*Remark.* For posterity, we note that the smoothness requirement (LS) *does not* imply that $\partial f(x)$ is a singleton.[3] The reason for this more general definition is that we want to concurrently treat problems with smooth and non-smooth objectives, and also feasible domains that are contained in lower-dimensional subspaces of $\mathcal{V}$. We also note that we will be mainly interested in cases where the above requirements all *fail* because $f$ and/or its derivatives blow up at the boundary of $\mathcal{X}$. By this token, we will not treat (LC)/(LS) as "blanket assumptions"; we discuss this in detail in the sequel.

**The oracle model.** From an algorithmic point of view, we aim to solve (Opt) by using iterative methods that require access to a *stochastic first-order oracle* (SFO) [37]. This means that, at each stage of the process, the optimizer can query a black-box mechanism that returns an estimate of the objective's gradient (or subgradient) at the queried point. Formally, when called at $x \in \mathcal{X}$, an SFO is assumed to return a random (dual) vector $g(x; \omega) \in \mathcal{V}^*$ where $\omega$ belongs to some (complete) probability space $(\Omega, \mathcal{F}, \mathbb{P})$. In practice, the oracle will be called repeatedly at a (possibly) random sequence of points $X_t \in \mathcal{X}$ generated by the algorithm under study. Thus, once $X_t$ has been generated at stage $t$, the oracle draws an i.i.d. sample $\omega_t \in \Omega$ and returns the dual vector:

$$g_t \equiv g(X_t; \omega_t) = \nabla f(X_t) + U_t \tag{1}$$

with $U_t \in \mathcal{V}^*$ denoting the "measurement error" relative to some selection $\nabla f(X_t)$ of $\partial f(X_t)$. In terms of measurability, we will write $\mathcal{F}_t$ for the history (natural filtration) of $X_t$; in particular, $X_t$ is $\mathcal{F}_t$-adapted, but $\omega_t$, $g_t$ and $U_t$ are not. Finally, we will also make the statistical assumption that

$$\mathbb{E}[U_t \mid \mathcal{F}_t] = 0 \quad \text{and} \quad \|U_t\|_*^2 \leq \sigma^2 \quad \text{for all } t = 1, 2, \ldots \tag{SFO}$$

where $\|\cdot\|_*$ denotes the dual norm on $\mathcal{V}^*$. This assumption is standard in the analysis of parameter-agnostic adaptive methods, cf. [6, 24, 26, 46] and references therein. For concreteness, we will refer to the case $\sigma = 0$ as deterministic – since, in that case, $U_t = 0$ for all $t$. Otherwise, if $\liminf_t \|U_t\|_* > 0$, the noise will be called *persistent* and the model will be called *stochastic*.

## 3 Relative Lipschitz continuity and smoothness

**3.1. Bregman functions.** We now proceed to describe a flexible template extending the standard Lipschitz continuity and Lipschitz smoothness conditions – (LC) and (LS) – to functions that are possibly singular at the boundary points of $\mathcal{X}$. The main idea of this extension revolves around the *non-Lipschitz* (NoLips) framework that was first studied by Birnbaum et al. [10] and then rediscovered independently by Bauschke et al. [7] and Lu et al. [29]. The key notion in this setting is that of a suitable "reference" *Bregman function*, which is assumed known to the optimizer, and which provides a geometry-adapted measure of divergence on $\mathcal{X}$. This is defined as follows:

**Definition 1.** A convex l.s.c. function $h\colon \mathcal{V} \to \mathbb{R} \cup \{\infty\}$ is a *Bregman function* on $\mathcal{X}$, if

1. $\operatorname{dom} \partial h \subseteq \mathcal{X}_\circ \subseteq \operatorname{dom} h$.

2. The subdifferential of $h$ admits a continuous selection $\nabla h(x) \in \partial h(x)$ for all $x \in \operatorname{dom} \partial h$.

3. $h$ is strongly convex, i.e., there exists some $K > 0$ such that

$$h(x') \geq h(x) + \langle \nabla h(x), x' - x \rangle + \frac{K}{2}\|x' - x\|^2 \tag{2}$$

for all $x \in \operatorname{dom} \partial h$, $x' \in \operatorname{dom} \partial h$.

---

[3]For example, the function $f\colon \mathbb{R}^2 \to \mathbb{R}$ with $f(x_1, 0) = x_1$ and $f(x_1, x_2) = \infty$ for $x_2 \neq 0$ is perfectly smooth on its domain ($x_2 = 0$); however, $\partial f(x_1, 0) = \{(1, v_2) : v_2 \in \mathbb{R}\}$, and this set is never a singleton.

The induced *Bregman divergence* of $h$ is then defined for all $x \in \operatorname{dom} \partial h$, $x' \in \operatorname{dom} h$ as

$$D(x', x) = h(x') - h(x) - \langle \nabla h(x), x' - x \rangle. \tag{3}$$

*Remark.* The notion of a Bregman function was first introduced by Bregman [13]. Our definition follows [5, 23, 35, 38] and leads to the smoothest presentation, but there are variant definitions where $h$ is not necessarily assumed strongly convex, cf. [8, 15] and references therein.

Some standard examples of Bregman functions are as follows:

- **Euclidean regularizer:** Let $\mathcal{X}$ be a convex subset of $\mathbb{R}^d$ endowed with the Euclidean norm $\|\cdot\|_2$. Then, the *Euclidean regularizer* on $\mathcal{X}$ is defined as $h(x) = \|x\|_2^2/2$ and the induced Bregman divergence is the standard square distance $D(x', x) = \|x' - x\|_2^2$ for all $x, x' \in \mathcal{X}$

- **Entropic regularizer:** Let $\mathcal{X} = \{x \in \mathbb{R}_+^d : \sum_{i=1}^d x_i = 1\}$ be the unit simplex of $\mathbb{R}^d$ endowed with the $L^1$-norm $\|\cdot\|_1$. Then, the *entropic regularizer* on $\mathcal{X}$ is $h(x) = \sum_i x_i \log x_i$ and the induced divergence is the relative entropy $D(x', x) = \sum_i x_i' \log(x_i'/x_i)$ for all $x' \in \mathcal{X}$ $x \in \operatorname{ri} \mathcal{X}$.

- **Log-barrier:** Let $\mathcal{X} = \mathbb{R}_{++}^d$ denote the (open) positive orthant of $\mathbb{R}^d$. Then, the *log-barrier* on $\mathcal{X}$ is defined as $h(x) = -\sum_{i=1}^d \log x_i$ for all $x \in \mathbb{R}_{++}^d$. The corresponding divergence is known as the *Itakura-Saito divergence* and is given by $D(x, x') = \sum_{i=1}^d (x_i/x_i' - \log(x_i/x_i') - 1)$ [15].

**3.2. Relative continuity.** With this background in hand, we proceed to discuss how to extend the Lipschitz regularity assumptions of Section 2 to account for problems with singular objective functions. We begin with the notion of *relative continuity* (RC), as introduced by Lu [28] and extended further in a recent paper by Zhou et al. [49]:

**Definition 2.** A convex l.s.c. function $f \colon \mathcal{V} \to \mathbb{R} \cup \{\infty\}$ is said to be *relatively continuous* if there exists some $G > 0$ such that

$$f(x) - f(x') \le \langle \nabla f(x), x - x' \rangle \le G\sqrt{2D(x', x)} \quad \text{for all } x' \in \operatorname{dom} h, x \in \operatorname{dom} \partial h. \tag{RC}$$

In the literature, there have been several extensions of (LC) to problems with singular objectives. Below we discuss some of these variants and how they can be integrated in the setting of Definition 2.

▶ **Example 1** (W$[f, h]$ continuity). This notion intends to single out sufficient conditions for the convergence of "proximal-like" methods like mirror descent. Specifically, following Teboulle [45], $f$ is said to be W$[f, h]$-continuous relative to $h$ on $\mathcal{X}$ (read: "$f$ is weakly $h$-continuous") if there exists some $G > 0$ such that, for all $t > 0$, we have

$$t\langle \nabla f(x), x - x' \rangle - D(x', x) \le \frac{t^2}{2} G^2 \quad \text{for all } x' \in \operatorname{dom} h, x \in \operatorname{dom} \partial h. \tag{W}$$

By rearranging the above quadratic polynomial in $t$, we note that its discriminant is $\Delta = [\langle \nabla f(x), x - x' \rangle]^2 - 2G^2 D(x', x)$, so it is immediate to check that (RC) holds.

▶ **Example 2** (Riemann–Lipschitz continuity). Concurrently to the above, Antonakopoulos et al. [2] introduced a Riemann-Lipschitz continuity condition extending (LC) as follows. Let $\|\cdot\|_x$ be a family of local norms on $\mathcal{X}$ (possibly induced by an appropriate Riemannian metric), and let $\|v\|_{x,*} = \max_{\|x'\|_x \le 1} \langle v, x' \rangle$ denote the corresponding dual norm. Then, $f$ is *Riemann–Lipschitz continuous* relative to $\|\cdot\|_x$ if there exists some $G > 0$ such that:

$$\|\nabla f(x)\|_{x,*} \le G \quad \text{for all } x \in \mathcal{X}. \tag{RLC}$$

As we show in the paper's supplement, (RLC) $\implies$ (RC) so (RC) is more general in this regard.

**3.3. Relative smoothness.** As discussed above, the notion of *relative smoothness* (RS) was introduced by [10] and independently rediscovered by [7, 29]. It is defined as follows:

**Definition 3.** A convex l.s.c. function $f \colon \mathcal{V} \to \mathbb{R} \cup \{\infty\}$ is said to be *relatively smooth* with respect to $h$ if

$$Lh - f \text{ is convex for some } L > 0. \tag{RS}$$

The main motivation behind this elegant definition is the following variational characterizations:

**Proposition 1.** *The following statements are equivalent for all $x, x' \in \operatorname{dom} \partial h$:*

1. *f satisfies* (RS).
2. *f satisfies the inequality* $f(x) \leq f(x') + \langle \nabla f(x'), x - x' \rangle + LD(x, x')$,
3. *f satisfies the inequality* $\langle \nabla f(x) - \nabla f(x'), x - x' \rangle \leq L[D(x, x') + D(x', x)]$.

A close variant of Proposition 1 appears in [7, 10, 29], so we do not prove it here. Instead, we discuss below a different extension of (LS) that turns out to be a special case of (RS).

▶ **Example 3** (Metric smoothness). Similar in spirit to (RLC), Antonakopoulos et al. [3] introduced an extension of (LS) that replaces the global norm $\|\cdot\|$ with a local norm $\|\cdot\|_x$, $x \in \mathcal{X}_\circ$. In particular, given such a norm, we say that $f$ is *metrically smooth* (relative to $\|\cdot\|_x$) if

$$\|\nabla f(x) - \nabla f(x')\|_{x,*} \leq L\|x - x'\|_{x'} \quad \text{for all } x, x' \in \operatorname{dom} \partial f. \tag{MS}$$

An observation that seems to have been overlooked by [3] is that (MS) $\implies$ (RS), so (RS) is more general. We prove this observation in the appendix.

**3.4. More examples.** Some concrete examples of problems satisfying (RC), (RS) or both (but not their Euclidean counterparts) are Fisher markets [10, 43], Poisson inverse problems [2, 7], support vector machines [28, 49], $D$-design [11, 29], etc. Because of space constraints, we do not detail these examples here; however, we provide an in-depth presentation of a Fisher market model in the appendix, along with a series of numerical experiments used to validate the analysis to come.

# 4 Adaptive mirror descent

We are now in a position to define the proposed *adaptive mirror descent* (ADAMIR) method. In abstract recursive form, ADAMIR follows the basic mirror descent template

$$x^+ = P_x(-\gamma g), \tag{MD}$$

where $P$ is a generalized Bregman proximal operator induced by $h$ (see below for the detailed definition), $g$ is a search direction determined by a (sub)gradient of $f$ at $x$, and $\gamma > 0$ is a step-size parameter. We discuss these elements in detail below, starting with the prox-mapping $P$.

**4.1. The prox-mapping.** Given a Bregman function $h$, its induced *prox-mapping* is defined as

$$P_x(v) = \arg\min_{x' \in \mathcal{X}}\{\langle v, x - x' \rangle + D(x', x)\} \quad \text{for all } x \in \operatorname{dom} \partial h, v \in \mathcal{V}^*, \tag{4}$$

where $D(x', x)$ denotes the Bregman divergence of $h$. Of course, in order for (4) to be well-defined, the $\arg\min$ must be attained in $\operatorname{dom} \partial h$. Indeed, we have:

**Proposition 2.** *The recursion* (MD) *satisfies* $x^+ \in \operatorname{dom} \partial h$ *for all* $x \in \operatorname{dom} \partial h$ *and all* $g \in \mathcal{V}^*$.

To streamline our discussion, we postpone the proof of Proposition 2 to the supplement. For now, we only note that it implies that the abstract recursion (MD) is *well-posed*, i.e., it can be iterated for all $t = 1, 2, \ldots$ to generate a sequence $X_t \in \mathcal{X}$.

**4.2. The method's step-size.** The next important element of (MD) is the method's step-size. In the unconstrained case, a popular adaptive choice is the so-called "inverse-sum-of-squares" policy

$$\gamma_t = 1 \Big/ \sqrt{\textstyle\sum_{s=1}^{t} \|\nabla f(X_s)\|_*^2}, \tag{5}$$

where $X_t$ is the series of iterates produced by the algorithm. However, in relatively continuous/smooth problems, this definition encounters two crucial issues. First, because the gradient of $f$ is unbounded (even over a bounded domain), the denominator of (5) may grow at an uncontrollable rate, leading to a step-size policy that vanishes too fast to be of any practical use. The second is that, if the problem is constrained, the extra terms entering the denominator of $\gamma_t$ do not vanish as the algorithm approaches a solution, so the (5) may still be unable to exploit the smoothness of the objective.

We begin by addressing the second issue. In the Euclidean case, the key observation is that the difference $\|x^+ - x\|$ must always vanish near a solution (even near the boundary), so we can use it as a proxy for $\nabla f(x)$ in constrained problems. This idea is formalized by the notion of the *gradient mapping* [37] that can be defined here as

$$\delta = \|x^+ - x\| \big/ \gamma. \tag{6}$$

On the other hand, in a Bregman setting, the prox-mapping tends to deflate gradient steps, so the norm difference between two successive iterates $x^+$ and $x$ of (MD) could be very small relative to the oracle signal that was used to generate the update. As a result, the Euclidean residual (6) could lead to a disproportionately large step-size that would be harmful for convergence. For this reason, we consider a gradient mapping that takes into account the Bregman geometry of the method and we set

$$\delta = \sqrt{D(x, x^+) + D(x^+, x)}\big/\gamma. \tag{7}$$

Obviously, when $h(x) = (1/2)\|x\|_2^2$, we readily recover the definition of the Euclidean gradient mapping (6). In general however, by the strong convexity of $h$, the value of this "Bregman residual" exceeds the corresponding Euclidean definition, so the induced step-size exhibits smoother variations that are more adapted to the framework in hand.

**4.3. The ADAMIR algorithm.** We are finally in a position to put everything together and define the *adaptive mirror descent* (ADAMIR) method. In this regard, combining the abstract template (MD) with the Bregman residual and "inverse-sum-of-squares" approach discussed above, we will consider the recursive policy

$$X_{t+1} = P_{X_t}(-\gamma_t g_t) \tag{8}$$

with $g_t$, $t = 1, 2, \ldots$, coming from a generic oracle model of the form (SFO), and with $\gamma_t$ defined as

$$\gamma_t = \frac{1}{\sqrt{\sum_{s=0}^{t-1} \delta_s^2}} \qquad \text{with } \delta_s^2 = \frac{D(X_s, X_{s+1}) + D(X_{s+1}, X_s)}{\gamma_s^2}. \tag{ADAMIR}$$

In the sequel, we will use the term "ADAMIR" to refer interchangeably to the update $X_t \leftarrow X_{t+1}$ and the specific step-size policy used within. The convergence properties of ADAMIR are discussed in detail in the next two sections (in both deterministic and stochastic problems); in the supplement, we also perform a numerical validation of the method in the context of a Fisher market model.

## 5 Deterministic analysis and results

We are now in a position to state our main convergence results for ADAMIR. We begin with the deterministic analysis ($\sigma = 0$), treating both the method's "time-average" as well as the induced trajectory of query points; the analysis for the stochastic case ($\sigma > 0$) is presented in the next section.

**5.1. Ergodic convergence and rate interpolation.** We begin with the convergence of the method's "time-averaged" state, i.e., $\bar{X}_T = (1/T) \sum_{t=1}^T X_t$.

**Theorem 1.** *Let $X_t$, $t = 1, 2, \ldots$, denote the sequence of iterates generated by ADAMIR, and let $D_1 = D(x^*, X_1)$. Then, ADAMIR simultaneously enjoys the following guarantees:*

*1. If $f$ satisfies (RC), we have:*

$$f(\bar{X}_T) - \min f \leq \frac{\sqrt{2}G\big[D_1 + 8G^2/\delta_0^2 + 2\log(1 + 2G^2T/\delta_0^2)\big]}{\sqrt{T}} + \frac{3\sqrt{2}G + 4G^2/\delta_0^2}{T}. \tag{9}$$

*2. If $f$ satisfies (RS), we have $f(\bar{X}_T) - \min f = \mathcal{O}(D_1/T)$.*

*3. If $f$ satisfies (RS) and (RC), we have:*

$$f(\bar{X}_T) - \min f \leq \left[f(X_1) - \min f + \left(2 + \frac{8G^2}{\delta_0^2} + 2\log\frac{4L^2}{\delta_0^2}\right)L\right]^2 \frac{D_1}{T}. \tag{10}$$

Theorem 1 shows that, up to logarithmic factors, ADAMIR achieves the min-max optimal bounds for functions in the $RC \cup RS$ oracle complexity class.[4] The starting point of the proof (which we detail in the supplement), is the following regret bound:

**Proposition 3.** *With notation as in Theorem 1, ADAMIR enjoys the regret bound*

$$\sum_{t=1}^T [f(X_t) - f(x^*)] \leq \frac{D_1}{\gamma_T} + \frac{\sum_{t=1}^T \gamma_t^2 \delta_t^2}{\gamma_T} + \sum_{t=1}^T \gamma_t \delta_t^2. \tag{11}$$

---

[4]We recall here that, in contrast to (LS), the $\mathcal{O}(1/T)$ rate is optimal in (RS), cf. Dragomir et al. [17].

The proof of Proposition 3 hinges on the specific definition of ADAMIR's step-size, and the exact functional form of the regret bound (11) plays a crucial role in the sequel. Specifically, under the regularity conditions (RC) and (RS), we respectively obtain the following key lemmas:

**Lemma 1.** *Under* (RC)*, the sequence of the Bregman residuals $\delta_t$ of* ADAMIR *is bounded as* $\delta_t^2 \leq 2G^2$ *for all* $t \geq 1$.

**Lemma 2.** *Under* (RS)*, the sequence of the Bregman residuals $\delta_t$ of* ADAMIR *is square-summable, i.e.,* $\sum_t \delta_t^2 < \infty$*. Consequently, the method's step-size converges to a strictly positive limit* $\gamma_\infty > 0$.

As we explain below, the boundedness estimate of Lemma 1 is necessary to show that the iterates of the method do not explode; however, without further assumptions, it is not possible to sharpen this bound.[5] The principal technical difficulty – and an important novelty of our analysis – is the stabilization of the step-size to a strictly positive limit in Lemma 2. This property of ADAMIR plays a crucial role because the method is not slowed down near a solution. To the best of our knowledge, there is no comparable result for the step-size of parameter-agnostic methods in the literature.[6]

Armed with these two lemmas, we obtain the following series of estimates:

1. Under (RC), the terms in the RHS of (11) can be bounded respectively as $\mathcal{O}(G\sqrt{T})$, $\mathcal{O}(\log(G^2 T)\sqrt{T})$, and $\mathcal{O}(G\sqrt{T})$. As a result, we obtain an $\tilde{\mathcal{O}}(1/\sqrt{T})$ rate of convergence.

2. Under (RS), all terms in the RHS of (11) can be bounded as $\mathcal{O}(1)$, so we obtain an $\mathcal{O}(1/T)$ convergence rate for $\bar{X}_T$.

For the details of these calculations (including the explicit constants and logarithmic terms that appear in the statement of Theorem 1), we refer the reader to the supplement.

**5.2. Other modes of convergence.** In complement to the analysis above, we provide below a spinoff result for the method's "last iterate", i.e., the actual trajectory of queried points. The formal statement is as follows.

**Theorem 2.** *Suppose that $f$ satisfies* (RC) *or* (RS)*. Then $X_t$ converges to* $\arg\min f$.

The main idea of the proof (which we detail in the appendix) consists of two steps. The first key step is to show that, under (RC) $\cup$ (RS), the iterates of ADAMIR have $\liminf f(X_t) = \min f$; we show this in Proposition C.1. Now, given the existence of a convergent subsequence, the rest of our proof strategy branches out depending on whether $f$ satisfies (RC) or (RS). Under (RS), the analysis relies on arguments that involve a quasi-Fejér argument as in [12, 16]. However, under (RC), the quasi-Fejér property fails, so we prove the convergence of $X_t$ via a novel induction argument that shows that the method's iterates remain trapped within a Bregman neighborhood of $x^*$ if they enter it with a sufficiently small step-size; we provide the relevant details in the supplement.

**Non-convex objectives.** We close this section with two remarks on non-convex objectives. First, Theorem 2 applies verbatim to non-convex functions satisfying the "secant inequality" [12, 32, 33, 50]

$$\inf\{\langle \nabla f(x), x - x^* \rangle : x^* \in \arg\min f, x \in \mathcal{K}\} > 0 \tag{SI}$$

for every closed subset $\mathcal{K}$ of $\mathcal{X}$ that is separated by neighborhoods from $\arg\min f$. In the supplement, our results have all been derived based on this more general condition (it is straightforward to verify that (SI) always holds for convex functions).

Even more generally, Lemma 2 also allows us to derive results for general non-convex problems. Indeed, the proof of Proposition 1 shows that $\min_{1 \leq t \leq T} \delta_t^2 = \mathcal{O}(1/T)$ *without* requiring any

---

[5]The only comparable result that we are aware of in the literature is Lemma 3.2 of [6] which, however, concerns a mirror-prox algorithm with two gradient queries per iteration, functions with bounded gradients, and a step-size defined in terms of a global norm. The specific bound of Lemma 1 is only possible thanks to the exact form of $\gamma_t$: since the gradient of $f$ can become arbitrarily large in terms of global norms, bounding $\delta_t$ would not have been possible otherwise.

[6]In more detail, Levy et al. [26], Li and Orabona [27] and Kavis et al. [24] establish the summability of a suitable residual sequence to sharpen the $\mathcal{O}(1/\sqrt{T})$ rate in their respective contexts, but this does not translate to a step-size stabilization result. Under (RC)/(RS), controlling the method's step-size is of vital importance because the gradients that enter the algorithm may be unbounded even over a bounded domain; this crucial difficulty does not arise in any of the previous works on adaptive methods for ordinary Lipschitz problems.

properties on $f$ other than relative smoothness. As a result, we conclude that the "best iterate" of the method – i.e., the iterate with the least residual – decays as $\mathcal{O}(1/T)$. This fact partially generalizes a similar result obtained in [27, 46] for ADAGRAD applied to non-convex problems; however, an in-depth discussion of this property would take us too far afield, so we do not attempt it.

## 6 Stochastic analysis

In this last section, we focus on the stochastic case ($\sigma > 0$). Our main results here are as follows.

**Theorem 3.** *Let $X_t$, $t = 1, 2, \ldots$, denote the sequence of iterates generated by* ADAMIR, *and let $D_1 = D(x^*, X_1)$ and $G_\sigma = G + \sigma/\sqrt{K}$. Then, under* (RC), *we have*

$$\mathbb{E}\left[f(\bar{X}_T) - f(x^*)\right] \leq (D_1 + H)\sqrt{\frac{\delta_0^2 + 2G_\sigma^2}{T}} \tag{12}$$

*where $H = 8G_\sigma^2/\delta_0^2 + 2\log(1 + 2G_\sigma^2 T/\delta_0^2)$.*

Finally, if (RS) kicks in, we have the sharper guarantee:

**Theorem 4.** *With notation as above, if $f$ satisfies* (RS), ADAMIR *enjoys the bound*

$$\mathbb{E}[f(\bar{X}_T) - f(x^*)] \leq (2 + D_1 + H)\left[\frac{A}{T} + \frac{B\sigma}{\sqrt{T}}\right] \tag{13}$$

*where:*

a)  $A = \delta_0 + 2[f(X_1) - \min f] + L\left(2 + 8G_\sigma^2/\delta_0^2 + 2\log(4L^2/\delta_0^2)\right).$  (14a)

b)  $B = \sqrt{(4 + 2H)/K}.$  (14b)

The full proof of Theorems 3 and 4 is relegated to the supplement, but the key steps are as follows:

**Step 1:** We first show that, under (RC), the method's residuals are bounded as $\delta_t^2 \leq 2G_\sigma^2$ (a.s.).

**Step 2:** With this at hand, the workhorse for our analysis is the following boxing bound for the mean "weighted" regret $\sum_{t=1}^T \mathbb{E}[\gamma_t\langle\nabla f(X_t), X_t - x^*\rangle]$:

$$\mathbb{E}\left[\gamma_T \sum_{t=1}^T [f(X_t) - f(x^*)]\right] \leq \mathbb{E}\left[\sum_{t=1}^T \gamma_t\langle\nabla f(X_t), X_t - x^*\rangle\right] \leq D_1 + \mathbb{E}\left[\sum_{t=1}^T \gamma_t^2\delta_t^2\right]$$

We prove this bound in the supplement, where we also show that $\mathbb{E}[\sum_{t=1}^T \gamma_t^2\delta_t^2] = \mathcal{O}(\log T)$.

At this point the analysis between Theorems 3 and 4 branches out. First, in the case of Theorem 3, we show that the method's step-size is bounded from below as $\gamma_t \geq 1/\sqrt{(\delta_0^2 + 2G_\sigma^2)t}$; the guarantee (12) then follows by the boxing bound. Instead, in the case of Theorem 4, the analysis is more involved and relies crucially on the lower bound $\gamma_t \geq 1/(A + B\sigma\sqrt{t})$. The bound (13) then follows by combining this lower bound for $\gamma_t$ with the regret boxing bound above.

In the supplement, we also conclude a series of numerical experiments in random Fisher markets that illustrate the method's adaptation properties in an archetypal non-Lipschitz problem.

## 7 Fisher markets: A case study

In this last section, we illustrate the convergence properties of ADAMIR in a Fisher equilibrium problem with linear utilities – both stochastic and deterministic. Following [40], a Fisher market consists of a set $\mathcal{N} = \{1, \ldots, n\}$ of $n$ *buyers* – or *players* – that seek to share a set $\mathcal{M} = \{1, \ldots, m\}$ of $m$ perfectly divisible goods (ad space, CPU/GPU runtime, bandwidth, etc.). The allocation mechanism for these goods follows a proportionally fair price-setting rule that is sometimes referred to as a *Kelly auction* [25]: each player $i = 1, \ldots, n$ bids $x_{ik}$ per unit of the $k$-th good, up the player's individual budget; for the sake of simplicity, we assume that this budget is equal to 1 for all players, so $\sum_{k=1}^m x_{ik} \leq 1$ for all $i = 1, \ldots, n$. The price of the $k$-th good is then set to be the sum of the players' bids, i.e., $p_k = \sum_{i\in\mathcal{N}} x_{ik}$; then, each player gets a prorated fraction of each good, namely $w_{ik} = x_{ik}/p_k$.

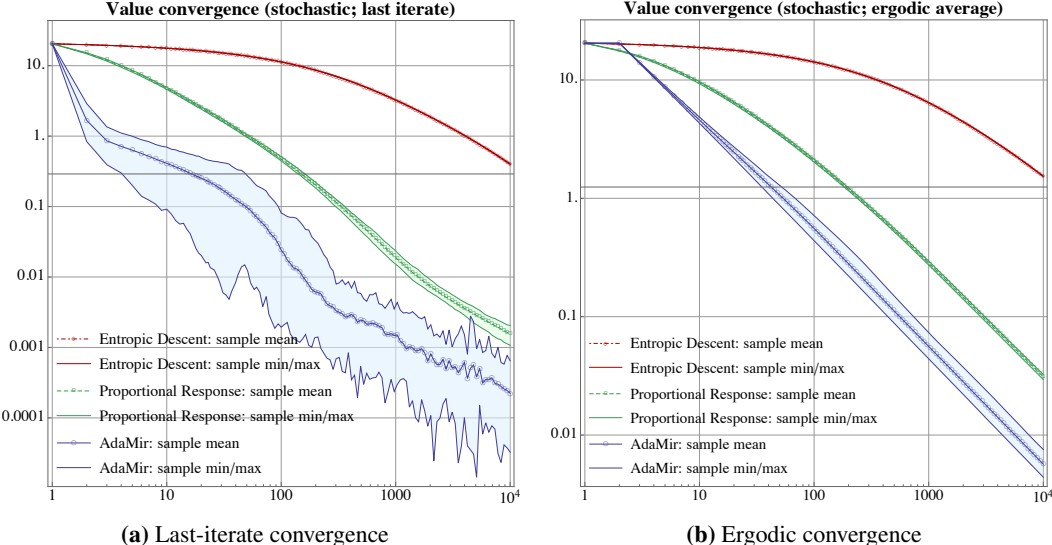

**(a)** Last-iterate convergence

**(b)** Ergodic convergence

**Figure 1:** Statistics for the convergence speed of entropic gradient descent, the proportional response algorithm and ADAMIR in a stochastic Fisher market with marginal utilities drawn i.i.d. at each epoch. The marked lines are the observed means from $S = 50$ realizations, whereas the shaded areas represent a 95% confidence interval.

As was observed by Shmyrev [43] (and discussed in more detail in the paper's supplement), the equilibrium points of a Fisher market can be obtained by solving the convex problem

$$
\begin{aligned}
\text{minimize} \quad & F(x;\theta) \equiv \sum_{k \in \mathcal{M}} p_k \log p_k - \sum_{i \in \mathcal{N}} \sum_{k \in \mathcal{M}} x_{ik} \log \theta_{ik} \\
\text{subject to} \quad & p_k = \sum_{i \in \mathcal{N}} x_{ik}, \ \sum_{k \in \mathcal{M}} x_{ik} = 1, \ \text{and } x_{ik} \geq 0 \text{ for all } k \in \mathcal{M}, i \in \mathcal{N},
\end{aligned}
\tag{Opt}
$$

where $\theta_{ik}$ denotes the marginal utility of the $i$-th player per unit of the $k$-th good. Accordingly, if these utilities fluctuate stochastically over time, the corresponding optimization problem involves the *mean* objective

$$
f(x) = \mathbb{E}[F(x;\omega)].
\tag{15}
$$

Because of the logarithmic terms involved, $F$ (and, a fortiori, $f$) cannot be Lipschitz continuous or smooth in the standard sense. However, as was shown by Birnbaum et al. [10], the problem satisfies (RS) over $\mathcal{X} = \{x \in \mathbb{R}_+^{nm} : \sum_{k \in \mathcal{M}} x_{ik} = 1\}$ relative to the negative entropy function $h(x) = \sum_{ik} x_{ik} \log x_{ik}$. As a result, mirror descent methods based on this Bregman function are natural candidates for solving (15).

In Fig. 1, we report the performance of the (non-adaptive) entropic gradient descent and proportional response algorithms studied by Birnbaum et al. [10], and we compare it to the performance of ADAMIR, which consistently outperforms both methods, in terms of both last-iterate and ergodic value convergence rates. We provide a more detailed analysis in the paper's supplement.

## 8 Concluding remarks

Our theoretical analysis confirms that ADAMIR concurrently achieves optimal rates of convergence in relatively continuous and relatively smooth problems, both stochastic or deterministic, constrained or unconstrained, and without requiring any prior knowledge of the problem's smoothness/continuity parameters. These appealing properties open the door to several future research directions, especially regarding the method's convergence properties in non-convex problems. The "best-iterate" discussion of Section 5 is a first step along the way, but many questions and problems remain open in this direction, especially regarding the convergence of the method's "last iterate" in stochastic, non-convex settings. We defer these questions to future work.

## Acknowledgments and Disclosure of Funding

This research was partially supported by the COST Action CA16228 "European Network for Game Theory" (GAMENET) and the French National Research Agency (ANR) in the framework of the "Investissements d'avenir" program (ANR-15-IDEX-02), the LabEx PERSYVAL (ANR-11-LABX-0025-01), MIAI@Grenoble Alpes (ANR-19-P3IA-0003), and the grant ALIAS (ANR-19-CE48-0018-01).

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
