## A  Bregman regularizers and mirror maps

Our goal in this appendix is to derive some basic properties for the class of Bregman proximal maps and mirror descent methods considered in the main body of our paper. Versions of the properties that we derive are known in the literature [see e.g., 8, 15, 21, 31, 38, and references therein].

To begin, we introduce two notions that will be particularly useful in the sequel. The first is the convex conjugate of a Bregman function $h$, i.e.,

$$h^*(y) = \max_{x \in \operatorname{dom} h} \{\langle y, x \rangle - h(x)\} \tag{A.1}$$

and the associated primal-dual *mirror map* $Q \colon \mathcal{V}^* \to \operatorname{dom} \partial h$:

$$Q(y) = \arg\max_{x \in \operatorname{dom} h} \{\langle y, x \rangle - h(x)\} \tag{A.2}$$

That the above is well-defined is a consequence of the fact that $h$ is proper, l.s.c., convex and coercive;[7] in addition, the fact that $Q$ takes values in $\operatorname{dom} \partial h$ follows from the fact that any solution of (A.2) must necessarily have nonempty subdifferential (see below). For completeness, we also recall here the definition of the Bregman proximal mapping

$$P_x(v) = \arg\min_{x' \in \operatorname{dom} h} \{\langle v, x - x' \rangle + D(x', x)\} \tag{prox}$$

valid for all $x \in \operatorname{dom} \partial h$ and all $v \in \mathcal{V}^*$.

We then have the following basic lemma connecting the above notions:

**Lemma A.1.** *Let $h$ be a regularizer in the sense of Definition 1 with $K$-strong convexity modulus. Then, for all $x \in \operatorname{dom} \partial h$ and all $v, y \in \mathcal{V}^*$ we have:*

1. *$x = Q(y) \iff y \in \partial h(x)$.*

2. *$x^+ = P_x(v) \iff \nabla h(x) + v \in \partial h(x) \iff x^+ = Q(\nabla h(x) + v)$.*

3. *Finally, if $x = Q(y)$ and $p \in \mathcal{X}$, we get:*

$$\langle \nabla h(x), x - p \rangle \le \langle y, x - p \rangle. \tag{A.3}$$

*Proof.* For the first equivalence, note that $x$ solves (A.1) if and only if $0 \in y - \partial h(x)$ and hence if and only if $y \in \partial h(x)$. Working in the same spirit for the second equivalence, we get that $x^+$ solves (prox) if and only if $\nabla h(x) + v \in \partial h(x^+)$ and therefore if and only if $x^+ = Q(\nabla h(x) + v)$.

For our last claim, by a simple continuity argument, it is sufficient to show that the inequality holds for the relative interior $\operatorname{ri} \mathcal{X}$ of $\mathcal{X}$ (which, in particular, is contained in $\operatorname{dom} \partial h$). In order to show this, pick a base point $p \in \operatorname{ri} \mathcal{X}$, and let

$$\phi(t) = h(x + t(p - x)) - [h(x) + \langle y, t(p - x) \rangle] \quad \text{for all } t \in [0, 1]. \tag{A.4}$$

Since, $h$ is strongly convex and $y \in \partial h(x)$ due to the first equivalence, it follows that $\phi(t) \ge 0$ with equality if and only if $t = 0$. Since, $\psi(t) = \langle \nabla h(x + t(p - x)) - y, p - x \rangle$ is a continuous selection of subgradients of $\phi$ and both $\phi$ and $\psi$ are continuous over $[0, 1]$, it follows that $\phi$ is continuously differentiable with $\phi' = \psi$ on $[0, 1]$. Hence, with $\phi$ convex and $\phi(t) \ge 0 = \phi(0)$ for all $t \in [0, 1]$, we conclude that $\phi'(0) = \langle \nabla h(x) - y, p - x \rangle \ge 0$ and thus we obtain the result. ☐

As a corollary, we have:

*Proof of Proposition 2.* Our claim follows directly from a tandem application of items (1) and (2) in Lemma A.1. ☐

To proceed, the basic ingredient for establishing connections between Bregman proximal steps is a generalization of the rule of cosines which is known in the literature as the "three-point identity" [15]. This will be our main tool for deriving the main estimates for our analysis. Being more precise, we have the following lemma:

---

[7]The latter holds because $h$ is strongly convex relative to $\|\cdot\|_x$, and $\|\cdot\|_x$ has been tacitly assumed bounded from below by a multiple $\mu\|\cdot\|$ of $\|\cdot\|$.

**Lemma A.2.** *Let $h$ be a regularizer in the sense of [Definition 1](). Then, for all $p \in \operatorname{dom} h$ and all $x, x' \in \operatorname{dom} \partial h$, we have:*

$$D(p, x') = D(p, x) + D(x, x') + \langle \nabla h(x') - \nabla h(x), x - p \rangle. \tag{A.5}$$

*Proof.* By definition:

$$
\begin{aligned}
D(p, x') &= h(p) - h(x') - \langle \nabla h(x'), p - x' \rangle \\
D(p, x) &= h(p) - h(x) - \langle \nabla h(x), p - x \rangle \\
D(x, x') &= h(x) - h(x') - \langle \nabla h(x'), x - x' \rangle.
\end{aligned}
\tag{A.6}
$$

The lemma then follows by adding the two last lines and subtracting the first. $\qquad \square$

Thanks to the three-point identity, we obtain the following estimate for the Bregman divergence before and after a mirror descent step:

**Proposition A.1.** *Let $h$ be a regularizer in the sense of [Definition 1]() with strong convexity modulus $K > 0$. Fix some $p \in \operatorname{dom} h$ and let $x^+ = P_x(v)$ for some $x \in \operatorname{dom} \partial h$ and $v \in \mathcal{V}^*$. We then have:*

$$D(p, x^+) \leq D(p, x) - D(x^+, x) + \langle v, x^+ - p \rangle \tag{A.7}$$

*and*

$$D(p, x^+) \leq D(p, x) + D(x, x^+) - \langle v, x - p \rangle. \tag{A.8}$$

*Proof.* By the three-point identity established in [Lemma A.2](), we have:

$$D(p, x) = D(p, x^+) + D(x^+, x) + \langle \nabla h(x) - \nabla h(x^+), x^+ - p \rangle \tag{A.9}$$

Rearranging terms then yields:

$$D(p, x^+) = D(p, x) - D(x^+, x) + \langle \nabla h(x^+) - \nabla h(x), x^+ - p \rangle \tag{A.10}$$

By [(A.3)]() and the fact that $x^+ = P_x(v)$ so $\nabla h(x) + v \in \partial h(x^+)$, the first inequality follows; the second one is obtained similarly. $\qquad \square$

# B  Convergence analysis of ADAMIR

In this appendix, we will illustrate in detail the convergence analysis of ADAMIR, which we present in pseudocode form as [Algorithm 1]() below. For ease of presentation we shall divide our analysis, as in the main body of our paper, into two sections: the deterministic and the stochastic one.

---

**Algorithm 1:** Adaptive mirror descent (ADAMIR)

---

1: **Initialize** $X_0 \neq X_1 \in \operatorname{dom} \partial h$; set $\delta_0 = [D(X_0, X_1) + D(X_1, X_0)]^{1/2}$
2: **for** $t = 1, 2, \ldots, T - 1$ **do**
3:     set $\gamma_t = \left( \sum_{s=0}^{t-1} \delta_s^2 \right)^{-1/2}$                                            # step
4:     get $g_t \leftarrow g(X_t; \omega_t)$                                               # feedback
5:     set $X_{t+1} = P_{X_t}(-\gamma_t g_t)$                                   # Bregman step
6:     set $\delta_t = [D(X_t, X_{t+1}) + D(X_{t+1}, X_t)]^{1/2} / \gamma_t$           # Bregman residual
7: **end for**
8: **return** $\bar{X}_T \leftarrow (1/T) \sum_{t=1}^{T} X_t$                                  # candidate solution

---

**B.1. The deterministic case.** We begin with the proof of [Lemma 1]() which provides an upper bound to the Bregman residuals generated by ADAMIR:

*Proof of [Lemma 1]().* By the definition of the Bregman proximal step in [(MD)]() and [Proposition A.1](), we have:

$$
\begin{aligned}
D(X_t, X_{t+1}) + D(X_{t+1}, X_t) &= \langle \nabla h(X_t) - \nabla h(X_{t+1}), X_t - X_{t+1} \rangle \\
&\leq \gamma_t \langle g_t, X_t - X_{t+1} \rangle.
\end{aligned}
\tag{B.1}
$$

Hence, by applying the (RC) condition of the objective we get:

$$D(X_t, X_{t+1}) + D(X_{t+1}, X_t) \leq \gamma_t G \sqrt{2D(X_{t+1}, X_t)}$$
$$\leq \gamma_t G \sqrt{2\left[D(X_{t+1}, X_t) + D(X_t, X_{t+1})\right]} \tag{B.2}$$

We thus get:

$$D(X_t, X_{t+1}) + D(X_{t+1}, X_t) \leq 2\gamma_t^2 G^2. \tag{B.3}$$

Hence, by the definition (7) of $\delta_t^2$, we conclude that

$$\delta_t^2 = \frac{D(X_t, X_{t+1}) + D(X_{t+1}, X_t)}{\gamma_t^2} \leq 2G^2. \tag{B.4}$$
$$\square$$

*Proof of Lemma 2.* Since the adaptive step-size policy $\gamma_t$ is decreasing and bounded from below $(\gamma_t) \geq 0$ we get that its limit exist,i.e.,

$$\lim_{t \to +\infty} \gamma_t = \gamma_\infty \quad \text{for some} \quad \gamma_\infty \geq 0 \tag{B.5}$$

Assume that $\gamma_\infty = 0$. By Proposition 1, we obtain:

$$f(X_{t+1}) \leq f(X_t) + \langle \nabla f(X_t), X_{t+1} - X_t \rangle + LD(X_{t+1}, X_t)$$
$$\leq f(X_t) - \frac{1}{\gamma_t} D(X_t, X_{t+1})$$
$$- \frac{1}{\gamma_t} D(X_{t+1}, X_t) + L\left[D(X_t, X_{t+1}) + D(X_{t+1}, X_t)\right] \tag{B.6}$$

whereas by recalling the definition of the residuals (ADAMIR) the above can be rewritten as follows:

$$f(X_{t+1}) \leq f(X_t) - \gamma_t \delta_t^2 + L\gamma_t^2 \delta_t^2 = f(X_t) - \frac{1}{2}\gamma_t \delta_t^2 - \frac{1}{2}\gamma_t \delta_t^2 + L\gamma_t^2 \delta_t^2 \tag{B.7}$$

Moreover, by rearranging and factorizing the common term $\gamma_t \delta_t^2$ we get:

$$\frac{1}{2}\gamma_t \delta_t^2 \leq f(X_t) - f(X_{t+1}) + \gamma_t \delta_t^2 \left[L\gamma_t - \frac{1}{2}\right] \tag{B.8}$$

Now, by combing that $\left[L\gamma_t - \frac{1}{2}\right] \leq 0$ for $\gamma_t \leq 1/2L$ and the fact that $\gamma_t$ converges to 0 by assumption, we get that there exists some $t_0 \in \mathbb{N}$ such that:

$$\left[L\gamma_t - \frac{1}{2}\right] \leq 0 \quad \text{for all } t > t_0 \tag{B.9}$$

Hence, by telescoping for $t = 1, 2, \ldots, T$ for sufficiently large $T$, we have

$$\frac{1}{2}\sum_{t=1}^{T} \gamma_t \delta_t^2 \leq f(X_1) - f(X_{T+1}) + \sum_{t=1}^{t_0} \left[L\gamma_t - \frac{1}{2}\right] \gamma_t \delta_t^2$$
$$\leq f(X_1) - \min_{x \in \mathcal{X}} f(x) + \sum_{t=1}^{t_0} \left[L\gamma_t - \frac{1}{2}\right] \gamma_t \delta_t^2 \tag{B.10}$$

Now, by applying the (LHS) of Lemma D.4 we get:

$$\frac{1}{2}\left[\frac{1}{\gamma_T} - \delta_0\right] \leq \frac{1}{2}\sqrt{\delta_0^2 + \sum_{t=1}^{T-1} \gamma_t \delta_t^2} \leq \sum_{t=1}^{T} \gamma_t \delta_t^2 \leq f(X_1) - \min_{x \in \mathcal{X}} f(x) + \sum_{t=1}^{t_0} \left[L\gamma_t - \frac{1}{2}\right] \gamma_t \delta_t^2 \tag{B.11}$$

Now, since $\gamma_t \to 0$ we get that $1/\gamma_t \to +\infty$ and hence the above yields that $+\infty \leq f(X_1) - \min_{x \in \mathcal{X}} f(x) + \sum_{t=1}^{t_0} \left[\frac{L}{K}\gamma_t - \frac{1}{2}\right] \gamma_t \delta_t^2$; a contradiction. Therefore we get that:

$$\lim_{t \to +\infty} \gamma_t = \gamma_\infty > 0 \tag{B.12}$$

Moreover, by recalling the definition of the adaptive step-size policy $\gamma_t$:

$$\gamma_t = \frac{1}{\sqrt{\delta_0^2 + \sum_{s=1}^{t-1}\delta_s^2}} \tag{B.13}$$

whereas after rearranging we obtain:

$$\sum_{s=1}^{t-1}\delta_s^2 = \frac{1}{\gamma_t^2} - \delta_0^2 \tag{B.14}$$

and therefore by taking limit on both sides we obtain:

$$\sum_{t=1}^{+\infty}\delta_t^2 = \lim_{t\to+\infty}\sum_{s=1}^{t-1}\delta_s^2 = \lim_{t\to+\infty}\frac{1}{\gamma_t^2} - \delta_0^2 = \frac{1}{\gamma_\infty^2} - \delta_0^2 < +\infty \tag{B.15}$$

and hence the result follows. $\qquad\square$

We proceed by providing an upper bound in terms of the Bregman divergence for the distance of the algorithm's iterates from a solution of (Opt):

**Lemma B.1.** *For all $x^* \in \mathcal{X}^*$, the iterates of Algorithm 1 satisfy the bound*

$$D(x^*, X_t) \leq D(x^*, X_1) + \sum_{s=1}^{T}\gamma_s^2\delta_s^2. \tag{B.16}$$

*Proof.* By the second part of Proposition A.1, we have:

$$\begin{aligned}
D(x^*, X_{s+1}) &\leq D(x^*, X_s) - \gamma_t\langle g_t, X_t - x^*\rangle + D(X_s, X_{s+1}) \\
&\leq D(x^*, X_s) + D(X_s, X_{s+1}) \\
&\leq D(x^*, X_s) + D(X_{s+1}, X_s) + D(X_s, X_{s+1})
\end{aligned} \tag{B.17}$$

Thus, by telescoping through $s = 1, 2, \ldots, t$, we obtain:

$$\begin{aligned}
D(x^*, X_t) &\leq D(x^*, X_1) + \sum_{s=1}^{t}[D(X_s, X_{s+1}) + D(X_{s+1}, X_s)] \\
&\leq D(x^*, X_1) + \sum_{s=1}^{T}[D(X_s, X_{s+1}) + D(X_{s+1}, X_s)] \\
&= D(x^*, X_1) + \sum_{s=1}^{T}\gamma_s^2\delta_s^2
\end{aligned} \tag{B.18}$$

where the last equality follows from the definition (7) of $\delta_t$. $\qquad\square$

With these intermediate results at our disposal, we are finally in a position to prove the core estimate (11) for ADAMIR:

*Proof of Proposition 3.* By the convexity of $f$, the definition of the Bregman proximal step in Algorithm 1 and Proposition A.1, we have:

$$f(X_t) - f(x^*) \leq \langle g_t, X_t - x^*\rangle \leq \frac{1}{\gamma_t}\langle \nabla h(X_t) - \nabla h(X_{t+1}), X_t - x^*\rangle. \tag{B.19}$$

Hence, by applying again the three-point identity (Lemma A.2), we obtain:

$$\begin{aligned}
f(X_t) - f(x^*) &\leq \frac{D(x^*, X_t) - D(x^*, X_{t+1})}{\gamma_t} + \frac{D(X_t, X_{t+1})}{\gamma_t} \\
&\leq \frac{D(x^*, X_t) - D(x^*, X_{t+1})}{\gamma_t} + \frac{D(X_t, X_{t+1}) + D(X_{t+1}, X_t)}{\gamma_t} \\
&= \frac{D(x^*, X_t) - D(x^*, X_{t+1})}{\gamma_t} + \gamma_t\delta_t^2
\end{aligned} \tag{B.20}$$

where the last equality follows readily from the definition (7) of $\delta_t$. Therefore, by summing through $t = 1, 2, \ldots, T$, we obtain:

$$\sum_{t=1}^{T} [f(X_t) - f(x^*)] \leq \frac{D(x^*, X_1)}{\gamma_1} + \sum_{t=2}^{T} \left[ \frac{1}{\gamma_t} - \frac{1}{\gamma_{t-1}} \right] D(x^*, X_t) + \sum_{t=1}^{T} \gamma_t \delta_t^2. \qquad \text{(B.21)}$$

Now, by Lemma B.1, the second term on the right-hand side (RHS) of (B.21) becomes:

$$
\begin{aligned}
\sum_{t=2}^{T} \left[ \frac{1}{\gamma_t} - \frac{1}{\gamma_{t-1}} \right] D(x^*, X_t) &\leq \sum_{t=2}^{T} \left[ \frac{1}{\gamma_t} - \frac{1}{\gamma_{t-1}} \right] \left( D(x^*, X_1) + \sum_{s=1}^{T} \gamma_s^2 \delta_s^2 \right) \\
&\leq \frac{D(x^*, X_1)}{\gamma_T} - \frac{D(x^*, X_1)}{\gamma_1} + \sum_{s=1}^{T} \gamma_s^2 \delta_s^2 \cdot \sum_{t=1}^{T} \left[ \frac{1}{\gamma_t} - \frac{1}{\gamma_{t-1}} \right] \\
&\leq \frac{D(x^*, X_1)}{\gamma_T} - \frac{D(x^*, X_1)}{\gamma_1} + \frac{\sum_{t=1}^{T} \gamma_t^2 \delta_t^2}{\gamma_T}. \qquad \text{(B.22)}
\end{aligned}
$$

Hence, by combining the above with (B.21), our claim follows. $\qquad \square$

With the regret bound (11) at our disposal, we may finally proceed with the proof of our main result concerning the universality of ADAMIR, i.e., Theorem 1 :

*Proof of Theorem 1.* Repeating the statement of Proposition 3, the iterate sequence $X_t$ generated by ADAMIR enjoys the bound:

$$\sum_{t=1}^{T} [f(X_t) - f(x^*)] \leq \frac{D(x^*, X_1)}{\gamma_T} + \frac{\sum_{t=1}^{T} \gamma_t^2 \delta_t^2}{\gamma_T} + \sum_{t=1}^{T} \gamma_t \delta_t^2 \qquad \text{(11)}$$

We now proceed to bound each term on the RHS of (11) from above. We consider three separate cases, first only under (RC), then under (RS) and finally when (RC) and (RS) holds.

**Case 1.** We begin with problems satisfying (RC).

- For the first term, Lemma 1 gives:

$$\frac{D(x^*, X_1)}{\gamma_T} = D(x^*, X_1) \sqrt{\sum_{t=0}^{T-1} \delta_t^2} \leq D(x^*, X_1) \sqrt{2 G^2 T}. \qquad \text{(B.23)}$$

- For the second term, we have:

$$\sum_{t=1}^{T} \gamma_t^2 \delta_t^2 \leq \sum_{t=1}^{T} \frac{\delta_t^2}{\sum_{s=0}^{t-1} \delta_s^2} = \sum_{t=1}^{T} \frac{\delta_t^2}{\delta_0^2 + \sum_{s=1}^{t-1} \delta_s^2}. \qquad \text{(B.24)}$$

Hence, by Lemmas 1 and D.5, we get:

$$
\begin{aligned}
\sum_{t=1}^{T} \gamma_t^2 \delta_t^2 &\leq 2 + \frac{8 G^2}{\delta_0^2} + 2 \log \left( 1 + \sum_{t=1}^{T-1} \frac{\delta_t^2}{\delta_0^2} \right) \\
&= 2 + \frac{8 G^2}{\delta_0^2} + 2 \log \left( \sum_{t=0}^{T-1} \frac{\delta_t^2}{\delta_0^2} \right) \\
&\leq 2 + \frac{8 G^2}{\delta_0^2} + 2 \log \frac{2 G^2 T}{\delta_0^2}. \qquad \text{(B.25)}
\end{aligned}
$$

- Finally, for the third term, we get:

$$\sum_{t=1}^{T} \gamma_t \delta_t^2 = \sum_{t=1}^{T} \frac{\delta_t^2}{\sqrt{\sum_{s=0}^{t-1} \delta_t^2}} = \sum_{t=1}^{T} \frac{\delta_t^2}{\sqrt{\delta_0^2 + \sum_{s=1}^{t-1} \delta_t^2}}. \qquad \text{(B.26)}$$

Hence, Lemmas 1 and D.4 again yield:

$$\sum_{t=1}^{T} \gamma_t \delta_t^2 \leq \frac{4G^2}{\delta_0} + 3\sqrt{2}G + 3\sqrt{\delta_0^2 + \sum_{t=1}^{T-1} \delta_t^2}$$

$$\leq \frac{4G^2}{\delta_0} + 3\sqrt{2}G + 3\sqrt{\sum_{t=0}^{T-1} \delta_t^2}$$

$$\leq \frac{4G^2}{\delta_0} + 3\sqrt{2}G + 3\sqrt{2G^2 T}. \tag{B.27}$$

The claim of Theorem 1 then follows by combining the above within the regret bound (11).

**Case 2.** We now turn to problems satisfying (RS). Recalling Lemma 2, we shall revisit the terms of (11). In particular, we have:

- For the first term, we have:

$$\frac{D(x^*, X_1)}{\gamma_T} = D(x^*, X_1)\sqrt{\sum_{t=0}^{T-1} \delta_t^2} \leq \frac{D(x^*, X_1)}{\gamma_\infty} \tag{B.28}$$

- For the second term, we have:

$$\sum_{t=1}^{T} \gamma_t^2 \delta_t^2 \leq \frac{1}{\delta_0^2} \sum_{t=1}^{T} \delta_t^2 \leq \frac{1}{\delta_0^2 \gamma_\infty^2} - 1 \tag{B.29}$$

- Finally, for the third term, we get:

$$\sum_{t=1}^{T} \gamma_t \delta_t^2 \leq \frac{1}{\delta_0} \sum_{t=1}^{T} \delta_t^2 \leq \frac{1}{\delta_0 \gamma_\infty^2} - \delta_0 \tag{B.30}$$

Combining all the above, the result follows.

**Case 3.** Finally, we consider objectives where (RC) and (RS) hold simultaneously. Now, by working in the same spirit as in the proof of Lemma 2 we get:

$$\frac{1}{2}\gamma_t \delta_t^2 \leq f(X_t) - f(X_{t+1}) + \gamma_t \delta_t^2 \left[ L\gamma_t - \frac{1}{2} \right] \tag{B.31}$$

which after telescoping $t = 1, \ldots, T$ it becomes:

$$\frac{1}{2} \sum_{t=1}^{T} \gamma_t \delta_t^2 \leq f(X_1) - \min_{x \in \mathcal{X}} f(x) + \sum_{t=1}^{T} \gamma_t \delta_t^2 \left[ L\gamma_t - \frac{1}{2} \right] \tag{B.32}$$

Now, after denoting:

$$t_0 = \max\{t \in \mathbb{N} : 1 \leq t \leq T \text{ such that } \gamma_t \geq \frac{1}{2L}\} \tag{B.33}$$

and decomposing the sum we get:

$$\frac{1}{2} \sum_{t=1}^{T} \gamma_t \delta_t^2 \leq f(X_1) - \min_{x \in \mathcal{X}} f(x) + \sum_{t=1}^{t_0} \gamma_t \delta_t^2 \left[ L\gamma_t - \frac{1}{2} \right] + \sum_{t=t_0+1}^{T} \gamma_t \delta_t^2 \left[ L\gamma_t - \frac{1}{2} \right]$$

$$\leq f(X_1) - \min_{x \in \mathcal{X}} f(x) + \sum_{t=1}^{t_0} \gamma_t \delta_t^2 \left[ L\gamma_t - \frac{1}{2} \right]$$

$$\leq f(X_1) - \min_{x \in \mathcal{X}} f(x) + L \sum_{t=1}^{t_0} \gamma_t^2 \delta_t^2 \tag{B.34}$$

On the other hand, by applying Lemma D.5, we have:

$$\sum_{t=1}^{t_0} \gamma_t^2 \delta_t^2 \leq 2 + \frac{8G^2}{\delta_0^2} + 2\log\left(1 + \sum_{t=1}^{t_0-1} \frac{\delta_t^2}{\delta_0^2}\right)$$

$$= 2 + \frac{8G^2}{\delta_0^2} + 2\log\left(\frac{1}{\delta_0^2}\left[\delta_0^2 + \sum_{t=1}^{t_0-1}\delta_t^2\right]\right)$$

$$= 2 + \frac{8G^2}{\delta_0^2} + 2\log\frac{1}{\delta_0^2\gamma_{t_0}^2} \tag{B.35}$$

and by definition of $t_0$ we get:

$$\sum_{t=1}^{t_0} \gamma_t^2 \delta_t^2 \leq 2 + \frac{8G^2}{\delta_0^2} + 2\log\frac{4L^2}{\delta_0^2}. \tag{B.36}$$

which yields:

$$\sum_{t=1}^{T} \gamma_t \delta_t^2 \leq f(X_1) - \min_{x\in\mathcal{X}} f(x) + L\left[2 + \frac{8G^2}{\delta_0^2} + 2\log\frac{4L^2}{\delta_0^2}\right] \tag{B.37}$$

The result then follows by plugging in the above bounds in (11). □

**B.2. The stochastic case.** In this appendix, we shall provide the stochastic part of our analysis. We start by providing an intermediate lemma concerning the class of (RC) objectives.

**Lemma B.2.** *Assume that $f$ satisfies (RC) and $X_t$ are the* ADAMIR *iterates run with feedback of the form* (SFO). *Then, the sequence of the residuals $\delta_t^2$ is bounded with probability* 1. *In particular, we have:*

$$\delta_t^2 \leq \tilde{G}^2 = \left[\sqrt{2}G + \sqrt{\frac{2}{K}}\sigma\right]^2 \quad \text{for all } t = 1, 2, \ldots \text{ almost surely} \tag{B.38}$$

*Proof.* By working in the same spirit, we get that:

$$D(X_t, X_{t+1}) + D(X_{t+1}, X_t) \leq \gamma_t \langle g_t, X_t - X_{t+1}\rangle \tag{B.39}$$

and by recalling that:

$$g_t = \nabla f(X_t) + U_t \tag{B.40}$$

we get with probability 1:

$$D(X_t, X_{t+1}) + D(X_{t+1}, X_t) \leq \gamma_t\left[\langle\nabla f(X_t), X_t - X_{t+1}\rangle + \langle U_t, X_t - X_{t+1}\rangle\right]$$

$$\leq \gamma_t\left[G\sqrt{2D(X_{t+1}, X_t)} + \|U_t\|_*\|X_t - X_{t+1}\|\right] \tag{B.41}$$

with the second inequality being obtained by (RC). Now, by invoking the strong convexity assumption of $K$, the (LHS) of the above becomes:

$$\gamma_t\left[G\sqrt{2D(X_{t+1}, X_t)} + \|U_t\|_*\|X_t - X_{t+1}\|\right] \leq \gamma_t[G\sqrt{2(D(X_{t+1}, X_t) + D(X_t, X_{t+1}))}$$

$$+ \|U_t\|_*\sqrt{\frac{2}{K}(D(X_{t+1}, X_t) + D(X_t, X_{t+1}))}] \tag{B.42}$$

which in turn yields:

$$D(X_t, X_{t+1}) + D(X_{t+1}, X_t) \leq \gamma_t\sqrt{D(X_{t+1}, X_t) + D(X_t, X_{t+1})}\left[\sqrt{2}G + \sqrt{\frac{2}{K}}\|U_t\|_*\right] \tag{B.43}$$

Therefore, we get:

$$D(X_t, X_{t+1}) + D(X_{t+1}, X_t) \leq \gamma_t^2\left[\sqrt{2}G + \sqrt{\frac{2}{K}}\|U_t\|_*\right]^2 \tag{B.44}$$

and by *stochastic first-order oracle* (SFO) we get with probability 1:

$$D(X_t, X_{t+1}) + D(X_{t+1}, X_t) \leq \gamma_t^2 \left[ \sqrt{2}G + \sqrt{\frac{2}{K}}\sigma \right]^2 \tag{B.45}$$

or equivalently,

$$\delta_t^2 = \frac{D(X_t, X_{t+1}) + D(X_{t+1}, X_t)}{\gamma_t^2} \leq \left[ \sqrt{2}G + \sqrt{\frac{2}{K}}\sigma \right]^2 \tag{B.46}$$

and the result follows. $\qquad\square$

Finally, we provide the proof of the first theorem for the stochastic setting.

*Proof of Theorem 3.* By the second part of Proposition A.1, we have:

$$\begin{aligned}
D(x^*, X_{t+1}) &\leq D(x^*, X_t) - \gamma_t\langle g_t, X_t - x^* \rangle + D(X_t, X_{t+1}) \\
&\leq D(x^*, X_t) - \gamma_t\langle g_t, X_t - x^* \rangle + D(X_{t+1}, X_t) + D(X_t, X_{t+1}) \\
&\leq D(x^*, X_t) - \gamma_t\langle g_t, X_t - x^* \rangle + \gamma_t^2 \delta_t^2 \tag{B.47}
\end{aligned}$$

which yields after rearranging and summing $t = 1, \ldots, T$:

$$\sum_{t=1}^{T} \gamma_t\langle g_t, X_t - x^* \rangle \leq D(x^*, X_1) + \sum_{t=1}^{T} \gamma_t^2 \delta_t^2 \tag{B.48}$$

and by recalling that $g_t = \nabla f(X_t) + U_t$ and taking expectations on both sides we get:

$$\mathbb{E}\left[ \sum_{t=1}^{T} \gamma_t\langle \nabla f(X_t), X_t - x^* \rangle \right] \leq D(x^*, X_1) + \mathbb{E}\left[ \sum_{t=1}^{T} \gamma_t\langle U_t, X_t - x^* \rangle \right] + \mathbb{E}\left[ \sum_{t=1}^{T} \gamma_t^2 \delta_t^2 \right] \tag{B.49}$$

First, we shall the (LHS) from below. In particular, we have by convexity:

$$\mathbb{E}\left[ \sum_{t=1}^{T} \gamma_t\langle \nabla f(X_t), X_t - x^* \rangle \right] \geq \mathbb{E}\left[ \sum_{t=1}^{T} \gamma_t(f(X_t) - f(x^*)) \right] \tag{B.50}$$

Moreover, by denoting $\tilde{G}^2 = \left[ \sqrt{2}G + \sqrt{\frac{2}{K}}\sigma \right]^2$ we have with probability 1:

$$\begin{aligned}
\sum_{t=1}^{T} \gamma_t(f(X_t) - f(x^*)) &= \sum_{t=1}^{T} \frac{1}{\sqrt{\delta_0^2 + \sum_{s=1}^{t-1}\delta_s^2}}(f(X_t) - f(x^*) \\
&\geq \sum_{t=1}^{T} \frac{1}{\sqrt{\delta_0^2 + \tilde{G}^2 t}}(f(X_t) - f(x^*)) \\
&\geq \sum_{t=1}^{T} \frac{1}{\sqrt{(\delta_0^2 + \tilde{G}^2)t}}(f(X_t) - f(x^*) \\
&\geq \frac{1}{\sqrt{(\delta_0^2 + \tilde{G}^2)T}} \sum_{t=1}^{T}(f(X_t) - f(x^*)) \tag{B.51}
\end{aligned}$$

with the second inequality being obtained by Lemma B.2. Hence, we get:

$$\mathbb{E}\left[ \sum_{t=1}^{T} \gamma_t\langle \nabla f(X_t), X_t - x^* \rangle \right] \geq \frac{1}{\sqrt{(\delta_0^2 + \tilde{G}^2)T}} \mathbb{E}\left[ \sum_{t=1}^{T}(f(X_t) - f(x^*)) \right] \tag{B.52}$$

We now turn our attention towards to the (LHS). In particular, we shall bound each term individually from above.

- For the term $\mathbb{E}\left[\sum_{t=1}^{T}\gamma_t\langle U_t, X_t - x^*\rangle\right]$:

$$\mathbb{E}\left[\sum_{t=1}^{T}\gamma_t\langle U_t, X_t - x^*\rangle\right] = \sum_{t=1}^{T}\mathbb{E}\left[\gamma_t\langle U_t, X_t - x^*\rangle\right]$$

$$= \sum_{t=1}^{T}\mathbb{E}\left[\mathbb{E}\left[\gamma_t\langle U_t, X_t - x^*\rangle|\mathcal{F}_t\right]\right]$$

$$= \sum_{t=1}^{T}\mathbb{E}\left[\gamma_t\mathbb{E}\left[\langle U_t, X_t - x^*\rangle|\mathcal{F}_t\right]\right]$$

$$= \sum_{t=1}^{T}\mathbb{E}\left[\gamma_t\langle\mathbb{E}[U_t|\mathcal{F}_t], X_t - x^*\rangle\right] = 0 \qquad \text{(B.53)}$$

with the third and the fourth equality being obtained by the fact that $\gamma_t$ and $X_t$ are $\mathcal{F}_t-$ measurable.

- For the term $\mathbb{E}\left[\sum_{t=1}^{T}\gamma_t^2\delta_t^2\right]$: By applying Lemma D.5 and Lemma B.2, we have with probability 1:

$$\sum_{t=1}^{T}\gamma_t^2\delta_t^2 \leq 2 + \frac{4\tilde{G}^2}{\delta_0^2} + 2\log(1 + \sum_{t=1}^{T}\frac{\delta_t^2}{\delta_0^2}) \leq 2 + \frac{4\tilde{G}^2}{\delta_0^2} + 2\log(1 + \frac{\tilde{G}^2}{K\delta_0^2}T) \qquad \text{(B.54)}$$

Therefore we get:

$$\mathbb{E}\left[\sum_{t=1}^{T}\gamma_t^2\delta_t^2\right] \leq 2 + \frac{4\tilde{G}^2}{\delta_0^2} + 2\log(1 + \frac{\tilde{G}^2}{\delta_0^2}T) \qquad \text{(B.55)}$$

Thus, combining all the above we obtain:

$$\frac{1}{\sqrt{(\delta_0^2 + \tilde{G}^2)T}}\mathbb{E}\left[\sum_{t=1}^{T}(f(X_t) - f(x^*))\right] \leq D(x^*, X_1) + 2 + \frac{4\tilde{G}^2}{\delta_0^2} + 2\log(1 + \frac{\tilde{G}^2}{\delta_0^2}T) \qquad \text{(B.56)}$$

and hence,

$$\mathbb{E}\left[\sum_{t=1}^{T}(f(X_t) - f(x^*))\right] \leq \sqrt{(\delta_0^2 + \tilde{G}^2)T}\left[D(x^*, X_1) + 2 + \frac{4\tilde{G}^2}{\delta_0^2} + 2\log(1 + \frac{\tilde{G}^2}{\delta_0^2})T)\right]$$

$$\text{(B.57)}$$

The result follows by dividing both sides by $T$. $\qquad\square$

*Proof of Theorem 4.* By Proposition 1, we have:

$$f(X_{t+1}) \leq f(X_t) + \langle\nabla f(X_t), X_{t+1} - X_t\rangle + LD(X_{t+1}, X_t)$$
$$\leq f(X_t) + \langle\nabla f(X_t), X_{t+1} - X_t\rangle + L\left[D(X_{t+1}, X_t) + D(X_t, X_{t+1})\right]$$
$$= f(X_t) + \langle g_t, X_{t+1} - X_t\rangle + \langle U_t, X_t - X_{t+1}\rangle + L\gamma_t^2\delta_t^2$$
$$\leq f(X_t) - \frac{1}{\gamma_t}\left[D(X_{t+1}, X_t) + D(X_t, X_{t+1})\right] + \|U_t\|_*\|X_t - X_{t+1}\| + L\gamma_t^2\delta_t^2$$
$$= f(X_t) - \gamma_t\delta_t^2 + \|U_t\|_*\|X_t - X_{t+1}\| + L\gamma_t^2\delta_t^2 \qquad \text{(B.58)}$$

Now, since $h$ is $K-$ strongly convex we have that:

$$\|X_t - X_{t+1}\| \leq \sqrt{\frac{2}{K}\left[D(X_{t+1}, X_t) + D(X_t, X_{t+1})\right]} = \sqrt{\frac{2}{K}}\gamma_t\delta_t \qquad \text{(B.59)}$$

and using the fact that the noise $\|U_t\|_* \leq \sigma$ almost surely, we have:

$$f(X_{t+1}) \leq f(X_t) - \gamma_t\delta_t^2 + \sqrt{\frac{2}{K}}\gamma_t\delta_t^2 + L\gamma_t^2\delta_t^2 \qquad \text{(B.60)}$$

Therefore, after rearranging and telescoping we get:

$$\sum_{t=1}^{T}\gamma_t\delta_t^2 \le 2\left[f(X_1) - \min_{x\in\mathcal{X}}f(x) + \sum_{t=1}^{T}\gamma_t\delta_t^2(L\gamma_t - \frac{1}{2}) + \sigma\sqrt{\frac{2}{K}}\sum_{t=1}^{T}\gamma_t\delta_t\right] \tag{B.61}$$

Now, let us bound each term of the (RHS) of the above individually:

- For the term $\sum_{t=1}^{T}\gamma_t\delta_t^2(L\gamma_t - \frac{1}{2})$ we first set:

$$t_0 = \max\{1 \le t \le T : \gamma_t \ge \frac{1}{2L}\} \tag{B.62}$$

Then, by decomposing the said sum we get:

$$\begin{aligned}
\sum_{t=1}^{T}\gamma_t\delta_t^2(L\gamma_t - \frac{1}{2}) &= \sum_{t=1}^{t_0}\gamma_t\delta_t^2(L\gamma_t - \frac{1}{2}) + \sum_{t=t_0+1}^{T}\gamma_t\delta_t^2(L\gamma_t - \frac{1}{2}) \\
&\le \sum_{t=1}^{t_0}\gamma_t\delta_t^2(L\gamma_t - \frac{1}{2}) \\
&\le L\sum_{t=1}^{t_0}\gamma_t^2\delta_t^2
\end{aligned} \tag{B.63}$$

with the second inequality being obtained by the definition of $t_0$. Now, due to the fact that $\delta_t^2 \le \tilde{G}^2$ almost surely (by invoking Lemma B.2) we have:

$$\begin{aligned}
L\sum_{t=1}^{t_0}\gamma_t^2\delta_t^2 &= L\sum_{t=1}^{t_0}\frac{\delta_t^2}{\delta_0^2 + \sum_{s=1}^{t-1}\delta_s^2} \\
&\le L\left[2 + \frac{4\tilde{G}^2}{\delta_0^2} + 2\log(1 + \frac{1}{\delta_0^2}\sum_{t=1}^{t_0-1}\delta_t^2)\right] \\
&\le L\left[2 + \frac{4\tilde{G}^2}{\delta_0^2} + 2\log\frac{1}{\delta_0^2}(\delta_0^2 + \sum_{t=1}^{t_0-1}\delta_t^2)\right] \\
&\le L\left[2 + \frac{4\tilde{G}^2}{\delta_0^2} + 2\log\frac{1}{\delta_0^2\gamma_{t_0}^2}\right]
\end{aligned} \tag{B.64}$$

Therefore, by the definition of $t_0$ we finally get with probability 1:

$$\sum_{t=1}^{T}\gamma_t\delta_t^2(L\gamma_t - \frac{1}{2}) \le L\left[2 + \frac{4\tilde{G}^2}{\delta_0^2} + 2\log\frac{4L^2}{\delta_0^2}\right] \tag{B.65}$$

- For the term $\sigma\sqrt{\frac{2}{K}}\sum_{t=1}^{T}\gamma_t\delta_t$ we have:

$$\sigma\sqrt{\frac{2}{K}}\sum_{t=1}^{T}\gamma_t\delta_t = \sigma\sqrt{\frac{2}{K}}\sum_{t=1}^{T}\sqrt{\gamma_t^2\delta_t^2} \le \sigma\sqrt{\frac{2}{K}}\sqrt{T}\sqrt{\sum_{t=1}^{T}\gamma_t^2\delta_t^2} \tag{B.66}$$

Therefore, by working in the same spirit as above we get:

$$\begin{aligned}
\sigma\sqrt{\frac{2}{K}}\sum_{t=1}^{T}\gamma_t\delta_t &\le \sigma\sqrt{\frac{2}{K}}\sqrt{2 + \frac{4\tilde{G}^2}{\delta_0^2} + 2\log(1 + \frac{1}{\delta_0^2}\sum_{t=1}^{T}\delta_t^2)} \\
&\le \sigma\sqrt{\frac{2}{K}}\sqrt{T}\sqrt{2 + \frac{4\tilde{G}^2}{\delta_0^2} + 2\log(1 + \frac{\tilde{G}^2}{\delta_0^2}T)}
\end{aligned} \tag{B.67}$$

On the other hand, we may the (LHS) from below as follows:

$$\sum_{t=1}^{T} \gamma_t \delta_t^2 \geq \gamma_T \sum_{t=1}^{T} \delta_t^2 \geq \gamma_T \left[ \delta_0^2 - \delta_0^2 + \sum_{t=1}^{T} \delta_t^2 \right] = \frac{\gamma_T}{\gamma_{T+1}^2} - \delta_0^2 \gamma_T = \frac{1}{\gamma_T} - \delta_0^2 \gamma_T \qquad \text{(B.68)}$$

So, combining the above:

$$\frac{1}{\gamma_T} - \delta_0^2 \gamma_T \leq 2(f(X_1) - \min_{x \in \mathcal{X}} f(x) + L \left[ 2 + \frac{4\tilde{G}^2}{\delta_0^2} + 2\log \frac{4L^2}{\delta_0^2} \right]$$
$$+ \sigma \sqrt{\frac{2}{K}} \sqrt{T} \sqrt{2 + \frac{4\tilde{G}^2}{\delta_0^2} + 2\log(1 + \frac{\tilde{G}^2}{\delta_0^2} T))} \quad \text{(B.69)}$$

which finally yields with probability 1:

$$\frac{1}{\gamma_T} \leq \delta_0 + 2(f(X_1) - \min_{x \in \mathcal{X}} f(x) + L \left[ 2 + \frac{4\tilde{G}^2}{\delta_0^2} + 2\log \frac{4L^2}{\delta_0^2} \right] + \sigma \sqrt{\frac{2}{K}} \sqrt{T} \sqrt{2 + \frac{4\tilde{G}^2}{\delta_0^2} + 2\log(1 + \frac{\tilde{G}^2}{\delta_0^2} T))}$$
$$\text{(B.70)}$$

and hence with probability 1:

$$\gamma_T \geq \left[ \delta_0 + 2(f(X_1) - \min_{x \in \mathcal{X}} f(x) + L \left[ 2 + \frac{4\tilde{G}^2}{\delta_0^2} + 2\log \frac{4L^2}{\delta_0^2} \right] + \sigma \sqrt{\frac{2}{K}} \sqrt{T} \sqrt{2 + \frac{4\tilde{G}^2}{\delta_0^2} + 2\log(1 + \frac{\tilde{G}^2}{\delta_0^2} T))} \right]^{-1}$$
$$\square$$

Therefore, by setting:

$$A = \delta_0 + 2(f(X_1) - \min_{x \in \mathcal{X}} f(x) + L \left[ 2 + \frac{4\tilde{G}^2}{\delta_0^2} + 2\log \frac{4L^2}{\delta_0^2} \right] \qquad \text{(B.71)}$$

and

$$B = \sigma \sqrt{\frac{2}{K}} \sqrt{2 + \frac{4\tilde{G}^2}{\delta_0^2} + 2\log(1 + \frac{\tilde{G}^2}{\delta_0^2} T))} \qquad \text{(B.72)}$$

we get that:

$$\mathbb{E} \left[ \sum_{t=1}^{T} (f(X_t) - f(x^*))\gamma_T \right] \geq \left( A + B\sqrt{T} \right)^{-1} \mathbb{E} \left[ \sum_{t=1}^{T} (f(X_t) - f(x^*)) \right] \qquad \text{(B.73)}$$

Moreover, working in the same spirit as in Theorem 3 we have:

$$\left( A + B\sqrt{T} \right)^{-1} \mathbb{E} \left[ \sum_{t=1}^{T} (f(X_t) - f(x^*)) \right] \leq \mathbb{E} \left[ \sum_{t=1}^{T} (f(X_t) - f(x^*))\gamma_T \right] \leq \left( D_1 + \mathbb{E} \left[ \sum_{t=1}^{T} \gamma_t^2 \delta_t^2 \right] \right)$$
$$\text{(B.74)}$$

which in turn yields:

$$\mathbb{E} \left[ \sum_{t=1}^{T} (f(X_t) - f(x^*)) \right] \leq \left( D_1 + \mathbb{E} \left[ \sum_{t=1}^{T} \gamma_t^2 \delta_t^2 \right] \right) \left( A + B\sqrt{T} \right) \qquad \text{(B.75)}$$

The result then follows by dividing both sides by $T$ and by the fact that $\mathbb{E} \left[ \sum_{t=1}^{T} \gamma_t^2 \delta_t^2 \right] = \mathcal{O}(\log T)$.

## C  Last iterate Convergence

Throughout this section we assume that $f$ satisfies the following weak-secant inequality of the form:

$$\inf\{\langle \nabla f(x), x - x^* \rangle : x^* \in \arg\min f, x \in \mathcal{K}\} > 0 \qquad \text{(SI)}$$

for every closed subset $\mathcal{K}$ of $\mathcal{X}$ that is separated by neighborhoods from $\arg\min f$. More precisely, our proof is divided in two parts. To begin with, we first show that under (RC) or (RS) the iterates of ADAMIR possess a convergent subsequence towards the solution set $\mathcal{X}^*$. Formally stated, we have the following proposition:

**Proposition C.1.** *Assume that $f$ is (RC) or (RS) and $X_t$ are the iterates generated by* ADAMIR. *Then, there exists a subsequence $X_{k_t}$ which converges to the solution set $\mathcal{X}^*$.*

*Proof.* Assume to the contrary that the sequence $X_t$ generated by ADAMIR admits no limit points in $\mathcal{X}^* = \arg\min f$. Then, there exists a (non-empty) closed set $\mathcal{K} \subseteq \mathcal{X}$ which is separated by neighbourhoods from $\arg\min f$ and is such that $X_t \in \mathcal{C}$ for all sufficiently large $t$. Then, by relabelling $X_t$ if necessary, we can assume without loss of generality that $X_t \in \mathcal{K}$ for all $t \in \mathbb{N}$. Thus, following the spirit of Lemma B.1, we have:

$$
\begin{aligned}
D(x^*, X_{t+1}) &\leq D(x^*, X_t) - \gamma_t \langle \nabla f(X_t), X_t - x^* \rangle + D(X_t, X_{t+1}) \\
&\leq D(x^*, X_t) - \gamma_t \langle \nabla f(X_t), X_t - x^* \rangle + [D(X_t, X_{t+1}) + D(X_{t+1}, X_t)] \\
&= D(x^*, X_t) - \gamma_t \langle \nabla f(X_t), X_t - x^* \rangle + \gamma_t^2 \delta_t^2
\end{aligned}
\tag{C.1}
$$

with the last equality being obtained by the definition of (7). Now, applying (SI) we get:

$$
D(x^*, X_{t+1}) \leq D(x^*, X_t) - \gamma_t \delta(\mathcal{K}) + \gamma_t^2 Z_t^2
\tag{C.2}
$$

with $\delta(\mathcal{K}) = \inf\{\langle \nabla f(x), x - x^* \rangle : x^* \in \arg\min f, x \in \mathcal{K}\} > 0$. Hence, by telescoping $t = 1, \ldots, T$, factorizing and setting $\beta_t = \sum_{t=1}^{T} \gamma_t$ we have:

$$
D(x^*, X_{T+1}) \leq D(x^*, X_1) - \beta_t \left[ \delta(\mathcal{K}) - \frac{\sum_{t=1}^{T} \gamma_t^2 Z_t^2}{\beta_t} \right]
\tag{C.3}
$$

Now, (C.3) will be the crucial lemma that will walk throughout our analysis. In particular, we will treat the different regularity conditions of (RC) and (RS) seperately.

**Case 1: The (RC) case.** Assume that $f$ satisfies (RC). By examining the asymptotic behaviour of each term individually, we obtain:

- For the term $\beta_T = \sum_{t=1}^{T} \gamma_t$, we have:

$$
\beta_T = \sum_{t=1}^{T} \frac{1}{\sqrt{\delta_0^2 + \sum_{j=1}^{t-1} \delta_t^2}} \geq \sum_{t=1}^{T} \frac{1}{\sqrt{\delta_0^2 + 2G^2 t}}
\tag{C.4}
$$

which yields that $\beta_T \to +\infty$ and more precisely $\beta_T = \Omega(\sqrt{T})$.

- For the term $\frac{\sum_{t=1}^{T} \gamma_t^2 \delta_t^2}{\beta_T}$, for the numerator we have:

$$
\begin{aligned}
\sum_{t=1}^{T} \gamma_t^2 \delta_t^2 &= \sum_{t=1}^{T} \frac{\delta_t^2}{\delta_0^2 + \sum_{j=1}^{t-1} \delta_j^2 / \delta_0^2} \\
&\leq 2 + 8G^2/\delta_0^2 + 2\log(1 + \sum_{t=1}^{T-1} \delta_t^2 / \delta_0^2) \\
&\leq 2 + 8G^2/\delta_0^2 + 2\log(1 + 2G^2 T/\delta_0^2)
\end{aligned}
\tag{C.5}
$$

which yields that $\sum_{t=1}^{T} \gamma_t^2 \delta_t^2 = \mathcal{O}(\log T)$, and combined with the fact that $\beta_t = \Omega(\sqrt{T})$ we readily get:

$$
\frac{\sum_{t=1}^{T} \gamma_t^2 \delta_t^2}{\beta_T} \to 0
\tag{C.6}
$$

So, combining all the above and letting $T \to +\infty$ in (C.3), we get that $D(x^*, X_{T+1}) \to -\infty$, a contradiction. Therefore, the result under (RC) follows.

**Case 2: The (RS) case.** On the other hand, assume that $f$ satisfies (RS). Recalling Lemma 2 and the fact that $\gamma_t$ is decreasing we have:

$$\sum_{t=1}^{T} \gamma_t \delta_t^2 \leq \sum_{t=1}^{+\infty} \delta_t^2 < +\infty \tag{C.7}$$

which by working as in Lemma 2 also yields:

$$\lim_{t \to +\infty} \gamma_t = \gamma_\infty > 0 \tag{C.8}$$

Additionally, since $\gamma_t$ is decreasing and bounded we also have that $\gamma_\infty = \inf_t \gamma_t$. Now, we shall re-examine the terms of (C.3). More precisely, we have:

- For $\beta_T$ we have:

$$\beta_T = \sum_{t=1}^{T} \gamma_t \geq \gamma_\infty \sum_{t=1}^{T} 1 = \gamma_\infty T \tag{C.9}$$

which in turn yields that $\beta_T \to +\infty$ and more precisely $\beta_T = \Omega(T)$.

- For the term $\frac{\sum_{t=1}^{T} \gamma_t^2 \delta_t^2}{\beta_T}$, for the numerator we have by the fact that $\gamma_t \leq 1/\delta_0$ and Lemma 2:

$$\sum_{t=1}^{T} \gamma_t \delta_t^2 \leq \frac{1}{\delta_0} \sum_{t=1}^{T} \delta_t^2 < +\infty \tag{C.10}$$

which yields that $\sum_{t=1}^{T} \gamma_t^2 \delta_t^2 = \mathcal{O}(1)$, which combined with (C.9) gives that:

$$\frac{\sum_{t=1}^{T} \gamma_t^2 \delta_t^2}{\beta_T} \to 0 \tag{C.11}$$

so, again combing the above and letting $T \to +\infty$ in (C.3), we get that $D(x^*, X_{T+1}) \to -\infty$, a contradiction. Therefore, the result follows also under (RS). □

Having all this at hand, we are finally in the position to prove the convergence of the actual iterates of the method. For that we will need an intermediate lemma that shall allow us to pass from a convergent subsequence to global convergence (see also [16], [41]).

**Lemma C.1.** *Let $\chi \in (0,1]$, $(\alpha_t)_{t\in\mathbb{N}}$, $(\beta_t)_{t\in\mathbb{N}}$ non-negative sequences and $(\varepsilon_t)_{t\in\mathbb{N}} \in l^1(\mathbb{N})$ such that $t = 1, 2, \ldots$:*

$$\alpha_{t+1} \leq \chi \alpha_t - \beta_t + \varepsilon_t \tag{C.12}$$

*Then, $\alpha_t$ converges.*

*Proof.* First, one shows that $\alpha_{t\in\mathbb{N}}$ is a bounded sequence. Indeed, one can derive directly that:

$$\alpha_{t+1} \leq \chi^{t+1} \alpha_0 + \sum_{k=0}^{t} \chi^{t-k} \varepsilon_k \tag{C.13}$$

Hence, $(\alpha_t)_{t\in\mathbb{N}}$ lies in $[0, \alpha_0 + \varepsilon]$, with $\varepsilon = \sum_{t=0}^{+\infty} \varepsilon_t$. Now, one is able to extract a convergent subsequence $(\alpha_{k_t})_{t\in\mathbb{N}}$, let say $\lim_{t\to+\infty} \alpha_{k_t} = \alpha \in [0, \alpha_0 + \varepsilon]$ and fix $\delta > 0$. Then, one can find some $t_0$ such that $\alpha_{k_{t_0}} - \alpha < \frac{\delta}{2}$ and $\sum_{m > t_{k_{t_0}}} \varepsilon_m < \frac{\delta}{2}$. That said, we have:

$$0 \leq \alpha_t \leq \alpha_{k_{t_0}} + \sum_{m > t_{k_{t_0}}} \varepsilon_m < \frac{\delta}{2} + \alpha + \frac{\delta}{2} = \alpha + \delta \tag{C.14}$$

Hence, $\limsup_t \alpha_t \leq \liminf_t \alpha_t + \delta$. Since, $\delta$ is chosen arbitrarily the result follows. □

*Proof of Theorem 2.* We will divide our proof in two parts by distinguishing the two different regularity cases.

**Case 1: The (RC) case.** Given that $\gamma_t$ is decreasing and bounded from below we have that its limit exists, denoted by $\gamma_\infty \geq 0$. We shall consider two cases:

1. $\gamma_\infty > 0$: Following the same reasoning with Lemma 2 we get that:

$$\sum_{t=1}^{T} \gamma_t^2 \delta_t^2 \leq \sum_{t=1}^{+\infty} \delta_t^2 < +\infty \tag{C.15}$$

Hence, by recalling the inequality:

$$D(x^*, X_{t+1}) \leq D(x^*, X_t) + \gamma_t^2 \delta_t^2 \ \text{ for all } \ x^* \in \mathcal{X}^* \tag{C.16}$$

whereas after taking infima on both sides with respect to $\mathcal{X}^*$, we get:

$$\inf_{x^* \in \mathcal{X}^*} D(x^*, X_{t+1}) \leq \inf_{x^* \in \mathcal{X}^*} D(x^*, X_t) + \gamma_t^2 \delta_t^2 \tag{C.17}$$

and since the sequence $\gamma_t^2 \delta_t^2$ is summable we can directly apply Lemma C.1 which yields that the sequence $\inf_{x^* \in \mathcal{X}^*} D(x^*, X_t)$ is convergent. Now, since by Proposition C.1, ADAMIR possesses a convergent subsequence towards the solution set $\mathcal{X}^*$ the result follows.

2. $\gamma_\infty = 0$: Pick some $\varepsilon > 0$ and consider the Bregman zone:

$$D_\varepsilon = \{x \in \mathcal{X} : D(\mathcal{X}^*, x) < \varepsilon\}. \tag{C.18}$$

Then, it suffices to show that $X_t \in D_\varepsilon$ for all sufficiently large $t$. In doing so, consider the inequality:

$$D(x^*, X_{t+1}) \leq D(x^*, X_t) - \gamma_t \langle \nabla f(X_t), X_t - x^* \rangle + \gamma_t^2 \delta_t^2$$
$$\leq D(x^*, X_t) - \gamma_t \langle \nabla f(X_t), X_t - x^* \rangle + \gamma_t^2 \frac{2G^2}{K} \tag{C.19}$$

with the second inequality being obtained by Lemma 1. To proceed, assume inductively that $X_t \in D_\varepsilon$. By the regularity assumptions of the regularizer $h$, it follows that there exists a $\delta-$neighbourhood contained in the closure of $D_{\varepsilon/2}$. So, by the (SI) condition we have:

$$\langle f(x), x - x^* \rangle \geq c > 0 \ \text{ for some } c \equiv c(\varepsilon) > 0 \text{ and for all } x \in D_\varepsilon \setminus D_{\varepsilon/2} \text{ and } x^* \in \mathcal{X}^* \tag{C.20}$$

We consider two cases:

- $X_t \in D_\varepsilon \setminus D_{\varepsilon/2}$: In. this case, we have:

$$D(x^*, X_{t+1}) \leq D(x^*, X_t) - \gamma_t \langle \nabla f(X_t), X_t - x^* \rangle + \gamma_t^2 \frac{2G^2}{K}$$
$$\leq D(x^*, X_t) - \gamma_t c + \gamma_t^2 \frac{2G^2}{K} \tag{C.21}$$

 Thus, provided that $\gamma_t \leq \frac{cK}{2G^2}$ we get that $D(x^*, X_{t+1}) \leq D(x^*, X_t)$. Hence, by taking infima on both sides relative to $x^* \in \mathcal{X}^*$, we get that $D(\mathcal{X}^*, X_{t+1}) \leq D(\mathcal{X}^*, X_t) < \varepsilon$.

- $X_t \in D_{\varepsilon/2}$: In this case, we have:

$$D(x^*, X_{t+1}) \leq D(x^*, X_t) - \gamma_t \langle \nabla f(X_t), X_t - x^* \rangle + \gamma_t^2 \frac{2G^2}{K}$$
$$\leq D(x^*, X_t) + \gamma_t^2 \frac{2G^2}{K} \tag{C.22}$$

 with the second inequality being obtained by the optimality of $x^*$. Now, provided that $\gamma_t^2 \leq \frac{\varepsilon K}{4G^2}$ or equivalently $\gamma_t \leq \frac{\sqrt{\varepsilon K}}{2G}$ we have:

$$D(x^*, X_{t+1}) \leq D(x^*, X_t) + \frac{\varepsilon}{2} \tag{C.23}$$

 whereas again by taking infima on both sides we get that $D(\mathcal{X}^*, X_{t+1}) \leq D(\mathcal{X}^*, X_t) + \frac{\varepsilon}{2} < \varepsilon$.

Hence, summarizing we have that $X_{t+1} \in D_\varepsilon$ whenever $X_t \in D_\varepsilon$ and $\gamma_t \leq \min\{\frac{cK}{2G^2}, \frac{\sqrt{\varepsilon K}}{2G}\}$. Hence, the result follows by Proposition C.1 and the fact that $\gamma_t \to 0$.

**Case 2: The** (RS) **case.** Recall that we have the following inequality,

$$D(x^*, X_{t+1}) \leq D(x^*, X_t) + \gamma_t^2 \delta_t^2 \ \text{ for all } \ x^* \in \mathcal{X}^* \tag{C.24}$$

whereas taking infima on both sides relative to $\mathcal{X}^*$ we readily get:

$$\inf_{x^* \in \mathcal{X}^*} D(x^*, X_{t+1}) \leq \inf_{x^* \in \mathcal{X}^*} D(x^*, X_t) + \gamma_t^2 \delta_t^2 \tag{C.25}$$

Now, by recalling that by Lemma 2, we have $\gamma_t^2 \delta_t^2$ is summable. we can apply directly Lemma C.1. Thus, we have the sequence $\inf_{x^* \in \mathcal{X}^*} D(x^*, X_t)$ is convergent. Moreover, Proposition C.1 guarantees that there a subsequence of $\inf_{x^* \in \mathcal{X}^*} \|X - x^*\|^2$ that converges to 0. We obtain that there exists also a subsequence of $\inf_{x^* \in \mathcal{X}^*} D(x^*, X_t)$ that converges to 0 and since $\inf_{x^* \in \mathcal{X}^*} D(x^*, X_t)$ is convergent, we readily get that:

$$\inf_{x^* \in \mathcal{X}^*} \|x^* - X_t\|^2 \leq \inf_{x^* \in \mathcal{X}^*} D(x^*, X_t) \to 0 \tag{C.26}$$

and the proof is complete. $\qquad\square$

# D   Lemmas on numerical sequences

In this appendix, we provide some necessary inequalities on numerical sequences that we require for the convergence rate analysis of the previous sections. Most of the lemmas presented below already exist in the literature, and go as far back as Auer et al. [4] and McMahan and Streeter [30]; when appropriate, we note next to each lemma the references with the statement closest to the precise version we are using in our analysis. These lemmas can also be proved by the general methodology outlined in Gaillard et al. [19, Lem. 14], so we only provide a proof for two ancillary results that would otherwise require some more menial bookkeeping.

**Lemma D.1** (30, 26). *For all non-negative numbers $\alpha_1, \ldots \alpha_t$, the following inequality holds:*

$$\sqrt{\sum_{t=1}^T \alpha_t} \leq \sum_{t=1}^T \frac{\alpha_t}{\sqrt{\sum_{i=1}^t \alpha_i}} \leq 2\sqrt{\sum_{t=1}^T \alpha_t} \tag{D.1}$$

**Lemma D.2** (26). *For all non-negative numbers $\alpha_1, \ldots \alpha_t$, the following inequality holds:*

$$\sum_{t=1}^T \frac{\alpha_t}{1 + \sum_{i=1}^t \alpha_i} \leq 1 + \log\left(1 + \sum_{t=1}^T \alpha_t\right) \tag{D.2}$$

**Lemma D.3.** *Let $b_1, \ldots, b_t$ a sequence of non-negative numbers with $b_1 > 0$. Then, the following inequality holds:*

$$\sum_{t=1}^T \frac{b_t}{\sum_{i=1}^t b_i} \leq 2 + \log\left(\frac{\sum_{t=1}^T b_t}{b_1}\right) \tag{D.3}$$

*Proof.* It is directly obtained by applying Lemma D.2 for the sequence $\alpha_t = b_t/b_1$. $\qquad\square$

The following set of inequalities are due to [6]. For completeness, we provide a sketch of their proof.

**Lemma D.4** (6). *For all non-negative numbers: $\alpha_1, \ldots \alpha_t \in [0, \alpha]$, $\alpha_0 \geq 0$, the following inequality holds:*

$$\sqrt{\alpha_0 + \sum_{t=1}^{T-1} \alpha_i} - \sqrt{\alpha_0} \leq \sum_{t=1}^T \frac{\alpha_t}{\sqrt{\alpha_0 + \sum_{i=1}^{t-1} \alpha_j}} \leq \frac{2\alpha}{\sqrt{\alpha_0}} + 3\sqrt{\alpha} + 3\sqrt{\alpha_0 + \sum_{t=1}^{T-1} \alpha_t} \tag{D.4}$$

**Lemma D.5.** *For all non-negative numbers: $\alpha_1, \ldots \alpha_t \in [0, \alpha]$, $\alpha_0 \geq 0$, we have:*

$$\sum_{t=1}^T \frac{\alpha_t}{\alpha_0 + \sum_{i=1}^{t-1} \alpha_i} \leq 2 + \frac{4\alpha}{\alpha_0} + 2\log\left(1 + \sum_{t=1}^{T-1} \frac{\alpha_t}{\alpha_0}\right) \tag{D.5}$$

*Proof.* Let us denote

$$T_0 = \min\{t \in [T] : \textstyle\sum_{j=1}^{t-1} \alpha_j \geq \alpha\} \tag{D.6}$$

Then, dividing the sum by $T_0$, we get:

$$\sum_{t=1}^{T} \frac{\alpha_t}{\alpha_0 + \sum_{i=1}^{t-1} \alpha_i} \leq \sum_{t=1}^{T_0-1} \frac{\alpha_t}{\alpha_0 + \sum_{i=1}^{t-1} \alpha_i} + \sum_{t=T_0}^{T} \frac{\alpha_t}{\alpha_0 + \sum_{i=1}^{t-1} \alpha_i}$$

$$\leq \frac{1}{\alpha_0} \sum_{t=1}^{T_0-1} \alpha_t + \sum_{t=T_0}^{T} \frac{\alpha_t}{1/2\alpha_0 + 1/2\alpha + 1/2 \sum_{j=1}^{t-1} \alpha_j}$$

$$\leq \frac{\alpha}{\alpha_0} + 2 \sum_{t=T_0}^{T} \frac{\alpha_i/\alpha_0}{1 + \sum_{j=T_0}^{t} \alpha_j/\alpha_0}$$

$$\leq \frac{2\alpha}{\alpha_0} + 2 + 2\log\left(1 + \sum_{t=T_0}^{T} \alpha_i/\alpha_0\right)$$

$$\leq \frac{2\alpha}{\alpha_0} + 2 + 2\log\left(1 + \sum_{t=1}^{T} \alpha_i/\alpha_0\right) \tag{D.7}$$

where we used the fact that $\sum_{j=1}^{T_0-2} \alpha_j \leq \alpha$ as well as for all $t \geq T_0$, $\sum_{j=1}^{t-1} \alpha_j \geq \alpha$ (both follow from the definition of $T_0$) and Lemma D.2. $\qquad\square$

# E  Fisher markets: A case study

**E.1. The Fisher market model.**  In this appendix, we provide some more details on the Fisher market model discussed in Section 7.

To begin, if the marginal utility of the $i$-th player per unit of the $k$-th good is $\theta_{ik}$, the agent's total utility will be

$$u_i(x_i; x_{-i}) = \sum_{k \in \mathcal{M}} \theta_{ik} w_{ik} = \sum_{k \in \mathcal{M}} \frac{\theta_{ik} x_{ik}}{\sum_{j \in \mathcal{N}} x_{jk}}, \tag{E.1}$$

where $x_i = (x_{ik})_{k \in \mathcal{M}}$ denotes the bid profile of the $i$-th player, and we use the shorthand $(x_i; x_{-i}) = (x_1, \ldots, x_i, \ldots, x_n)$. A *Fisher equilibrium* is then reached when the players' prices bids follow a profile $x^* = (x_1^*, \ldots, x_n^*)$ such that

$$u_i(x_i^*; x_{-i}^*) \geq u_i(x_i; x_{-i}^*) \tag{Eq}$$

for all $i \in \mathcal{N}$ and all $x_i = (x_{ik})_{k \in \mathcal{M}}$ such that $x_{ik} \geq 0$ and $\sum_{k \in \mathcal{M}} x_{ik} = 1$.[8]

As was observed by Shmyrev [43], the equilibrium problem (Eq) can be rewritten equivalently as

$$
\begin{aligned}
\text{minimize} \quad & F(x; \theta) \equiv \sum_{k \in \mathcal{M}} p_k \log p_k - \sum_{i \in \mathcal{N}} \sum_{k \in \mathcal{M}} x_{ik} \log \theta_{ik} \\
\text{subject to} \quad & p_k = \sum_{i \in \mathcal{N}} x_{ik}, \ \sum_{k \in \mathcal{M}} x_{ik} = 1, \ \text{and } x_{ik} \geq 0 \text{ for all } k \in \mathcal{M}, i \in \mathcal{N},
\end{aligned}
\tag{Opt}
$$

with the standard continuity convention $0 \log 0 = 0$. In the above, the agents' marginal utilities are implicitly assumed fixed throughout the duration of the game. On the other hand, if these utilities fluctuate stochastically over time, the corresponding reformulation instead involves the *mean* objective

$$f(x) = \mathbb{E}[F(x; \omega)]. \tag{E.2}$$

Because of the logarithmic terms involved, $F$ (and, a fortiori, $f$) cannot be Lipschitz continuous or smooth in the standard sense. However, as was shown by Birnbaum et al. [10], the problem satisfies (RS) over $\mathcal{X} = \{x \in \mathbb{R}_+^{nm} : \sum_{k \in \mathcal{M}} x_{ik} = 1\}$ relative to the negative entropy function $h(x) = \sum_{ik} x_{ik} \log x_{ik}$. As a result, mirror descent methods based on this Bregman function are natural candidates for solving (15).

---

[8]It is trivial to see that, in this market problem, all users would saturate their budget constraints at equilibrium, i.e., $\sum_{k \in \mathcal{M}} x_{ik} = 1$ for all $i \in \mathcal{N}$.

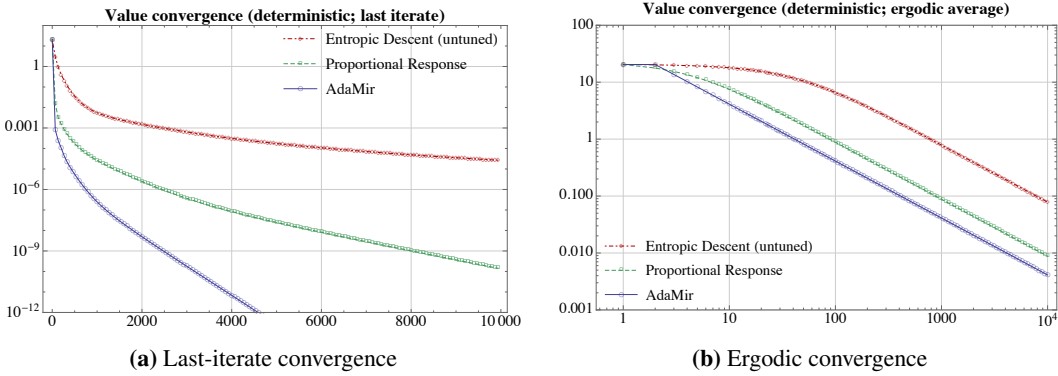

**(a)** Last-iterate convergence   **(b)** Ergodic convergence

**Figure 2:** The convergence speed of (EGD), (PR) and ADAMIR in a stationary Fisher market.

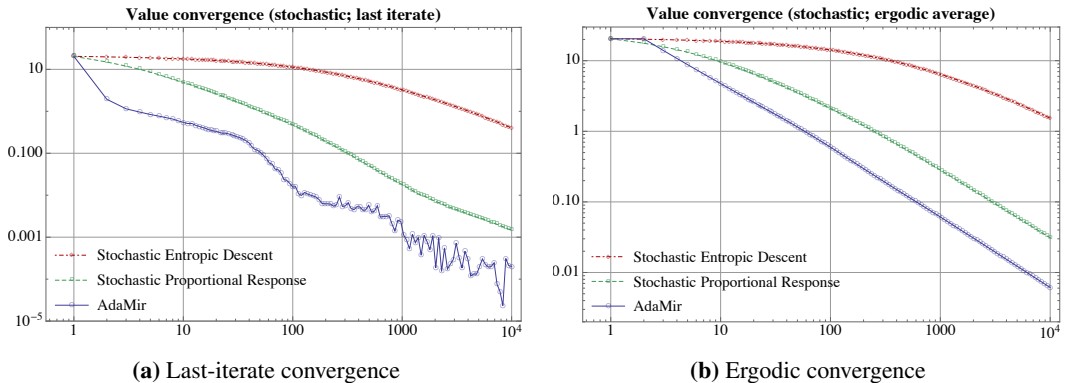

**(a)** Last-iterate convergence   **(b)** Ergodic convergence

**Figure 3:** The convergence speed of (EGD), (PR) and ADAMIR in a stochastic Fisher market, with marginal utilities drawn i.i.d. at each epoch.

In more detail, following standard arguments [8], the general mirror descent template (MD) relative to $h$ can be written as

$$x_{ik}^+ = \frac{x_{ik}\exp(-\gamma g_{ik})}{\sum_{l\in\mathcal{M}} x_{il}\exp(-\gamma g_{il})} \qquad (E.3)$$

where the (stochastic) gradient vector $g \equiv g(x;\theta)$ is given in components by

$$g_{ik} = 1 + \log p_k - \log\theta_{ik}. \qquad (E.4)$$

Explicitly, this leads to the entropic gradient descent algorithm

$$X_{ik,t+1} = \frac{X_{ik,t}(\theta_{ik}/p_k)^{\gamma_t}}{\sum_{l\in\mathcal{M}} X_{il,t}(\theta_{il}/p_l)^{\gamma_t}} \qquad (\text{EGD})$$

In particular, as a special case, the choice $\gamma = 1$ gives the *proportional response* (PR) algorithm of Wu and Zhang [47], namely

$$X_{ik,t+1} = \frac{\theta_{ik}w_{ik,t}}{\sum_{l\in\mathcal{M}} \theta_{il}w_{il,t}}, \qquad (\text{PR})$$

where $w_{ik,t} = X_{ik,t}\big/\sum_{j\in\mathcal{N}} X_{jk,t}$. As far as we aware, the PR algorithm is considered to be the most efficient method for solving *deterministic* Fisher equilibrium problems [10].

**E.2. Experimental validation and methodology.** For validation purposes, we ran a series of numerical experiments on a synthetic Fisher market model with $n = 50$ players sharing $m = 5$ goods, and utilities drawn uniformly at random from the interval $[2, 8]$. For stationary markets, the players' marginal utilities were drawn at the outset of the game and were kept fixed throughout; for stochastic models, the parameters were redrawn at each stage around the mean value of the stationary model (for consistency of comparisons). All experiments were run on a MacBook Pro with a 6-Core Intel i7 CPU clocking in at 2.6GHZ and 16 GB of DDR4 RAM at 2667 MHz.

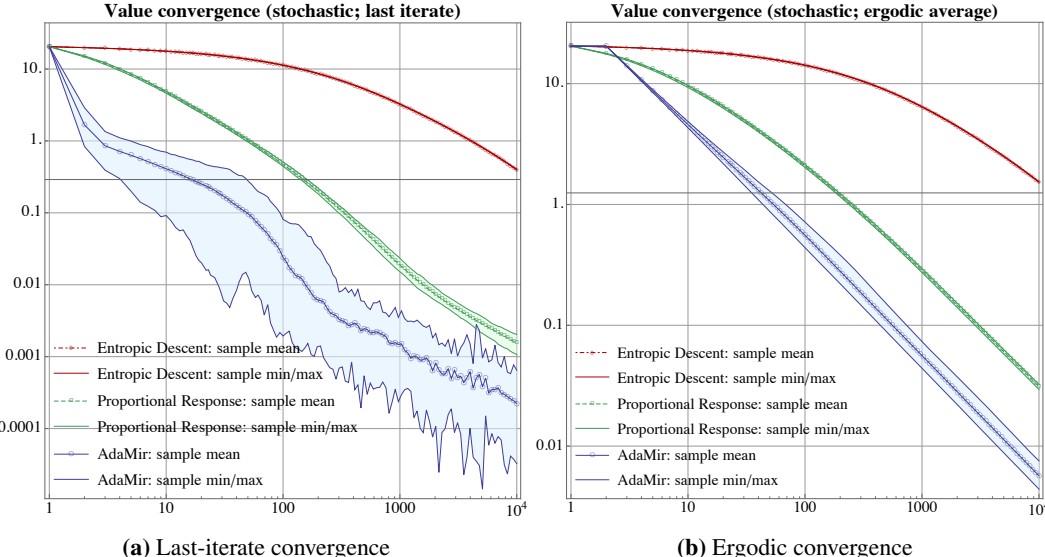

**(a)** Last-iterate convergence                      **(b)** Ergodic convergence

**Figure 4:** Statistics for the convergence speed of (EGD), (PR) and ADAMIR in a stochastic Fisher market, with marginal utilities drawn i.i.d. at each epoch. The marked lines are the observed means from $S = 50$ realizations, whereas the shaded areas represent a 95% confidence interval.

In each regime, we tested three algorithms, all initialized at the barycenter of $\mathcal{X}$: *a*) an untuned version of (EGD); *b*) the proportional response algorithm (PR); and *c*) ADAMIR. For stationary markets, we ran the untuned version of (EGD) with a step-size of $\gamma = .1$; (PR) was ran "as is", and ADAMIR was run with $\delta_0$ determined by drawing a second initial condition from $\mathcal{X}$. In the stochastic case, following the theory of Lu [28] and Antonakopoulos et al. [2], the updates of (EGD) and (PR) were modulated by a $\sqrt{t}$ factor to maintain convergence; by contrast, ADAMIR was run unchanged to test its adaptivity properties.

The results are reported in Figs. 2–4. For completeness, we plot the evolution of each method in terms of values of $f$, both for the "last iterate" $X_t$ and the "ergodic average" $\bar{X}_t$. The results for the deterministic case are presented in Fig. 2. For stochastic market models, we present a sample realization in Fig. 3, and a statistical study over $S = 50$ sample realizations in Fig. 4. In all cases, ADAMIR outperforms both (EGD) and (PR), in terms of both last-iterate and time-average guarantees.

An interesting observation is that each method's last iterate exhibits faster convergence than its time-average, and the convergence speed of the methods' time-averaged trajectories is faster than our worst-case predictions. This is due to the specific properties of the Fisher market model under consideration: more often than not, players tend to allocate all of their budget to a single good, so almost all of the problem's inequality constraints are saturated at equilibrium. Geometrically, this means that the problem's solution lies in a low-dimensional face of $\mathcal{X}$, which is identified at a very fast rate, hence the observed accelerated rate of convergence. However, this is a specificity of the market model under consideration and should not be extrapolated to other convex problems – or other market equilibrium models to boot.