# OpenReview forum: "Adaptive First-Order Methods Revisited: Convex Minimization without Lipschitz Requirements"
_NeurIPS.cc/2021/Conference — NeurIPS 2021 Poster_

### Official Review · Reviewer_xpRJ · 2021-07-13

**Rating:** 8
**Confidence:** 4

**Summary:**

This paper studies convex minimization problems lacking Lipschitz continuity of the objective and compactness of the feasible set. They assume access to a standard stochastic subgradient oracle with mean-zero noise and bounded variance. The main contribution of the paper is to propose an adaptive method based on Mirror descent and a clever step-size rule that attains optimal convergence rates under several regularity assumptions beyond standard Lipschitz continuity.

**Limitations And Societal Impact:**

The limitations were addressed. The paper studies a theoretical question with no direct societal impact.

**Main Review:**

The paper is well-written and easy to follow. It was a pleasure to read. The authors did a good job explaining the problem, the motivating limitations from the literature, and the connections to other methods. The use of examples and the clarifying remarks were very insightful.

The proposed method combines Mirror descent with a generalized version of the adaptive step-size rule popularized by AdaGrad. The method is simple and somewhat natural. The core innovation is the analysis of these algorithmic solutions under weaker conditions. The theoretical analysis appears to be mathematically sound. However, I did not check all the details.

Here are some points that the authors should address before publication:

- The column "Order-optimal" in Table 1 is slightly misleading. This column is not comparable among the rows because the classes of functions are different. Consider dropping the column or including the class explicitly.
- I would like to see how AdaMir performs against other methods in the literature; the paper would greatly benefit from a few numerical experiments.
- What does "partial" mean in Table 1?
- L98: adaptiving -> adapting.
- L121: Move the footnote to the previous sentence where you make the claim.
- L131: Delete one "sample."
- L134: Clarify this is the dual norm on V^*.
- L229: Typo double in the inline equation.
- L259: It is odd to present all the other statements explicitly and use the big-O notation for this one. Consider writing an explicit bound.
- L301: I cannot find (SI) defined anywhere. I am assuming you refer to (12)?

**Time Spent Reviewing:**

6

---

> ### Author Response · Authors · 2021-08-10
> **Replies to reviewer**
>
> We are sincerely grateful to the reviewer for their thoughtful review, their encouraging remarks and their strongly positive evaluation! We reply to the reviewer's questions below:
>
> 1. On the meaning of "order-optimal". We understand the reviewer's concern that the meaning of "order-optimal" could be ambiguous. We will break up the table accordingly to clearly distinguish between Lipschitz and non-Lipschitz results.
>
> 2. On a numerical evaluation of Adamir. We have already included a series of numerical experiments on Fisher markets in Appendix E. We would be happy to bring these results to the main body of the paper.
>
> 3. On the meaning of "partial" in Table 1. In the case of GMP, "partial" means that the algorithm does not interpolate between the relatively continuous and relatively smooth problem classes, even though it interpolates between  (cf. Lines 90-92 of the paper). In the case of AdaProx, this means that the conditions for which it guarantees convergence a subset of the RC/RS conditions (cf. Lines 95-96 of the paper). We will be happy to clarify this in the caption of Table 1 in the main body of the paper.
>
> 4. On the reference to (SI): yes, this was a broken reference that was supposed to point to (12), apologies for the confusion.
>
> 5. On more minor comments and typos: consider those fixed, thanks a lot for spotting them!
>
>
> We hope that the above answers your questions, and we will be sure to include these remarks in our revision. Thank you again for your thoughtful review and very positive evaluation!

---

> > ### Comment · Reviewer_xpRJ · 2021-08-17
> > **Reply to author**
> >
> > I thank the authors for their comments. I am mostly satisfied with the authors' reply. I didn't mention it in my original review, but I'd like the authors to comment on the originality of the proof techniques. Furthermore, I want the authors to compare their proofs with the ones in the literature. By emphasizing what is a simple extension of previous ideas and what is truly innovative.

---

> > > ### Author Response · Authors · 2021-08-19
> > > **On our proof techniques**
> > >
> > > Dear reviewer,
> > >
> > > Thanks a lot for reaching out, we are very happy for the opportunity to discuss in more detail our proofs and the difficulties we encountered - both technical and conceptual.
> > >
> > > In a nutshell, the principal challenges we faced stem from the fact that our paper simultaneously tackles several issues: functions with gradient singularities, (possibly constrained) domains with infinite Bregman diameter, and adaptivity with respect to the problem's regularity class and/or any randomness in the oracle. While some of these issues have already been tackled in the literature in isolation, they have not been treated  _all at once_: for example, the AcceleGrad and UnixGrad papers [22,23] achieve adaptivity but their proof techniques cannot treat domains with infinite Bregman diameter or gradient singularities; the analysis of AdaGrad in [24] can handle unconstrained problems when the function is Lipschitz smooth in the Euclidean sense and its smoothness modulus is known in advance; the cited papers [1], [27] and [45] treat online and stochastic problems with gradient singularities, but their proofs do not concern adaptive methods; etc.
> > >
> > > We mention this to clarify the following:
> > >
> > > 1. Treating all these issues in a synthetic manner requires more than the sum of existing proof techniques: as we explain in detail below, our paper contains several novel methodologies which we believe could be of future use in the analysis of adaptive methods.
> > > 2. Much of the difficulty of our proofs is hidden in the definition of the adaptive step-size policy $\gamma_t$. Specifically, the definition (7) of the (symmetrized) Bregman residual may appear natural _a posteriori_, but the fact that the standard gradient mapping (6) is not sufficient for our purposes can only be inferred _after_ trying to adapt existing proof techniques to the NoLips setting. In this regard, the originality of the proposed step-size policy cannot be overlooked: the definition of $\gamma_t$ is the starting point for our proofs, and it is not possible to assess the originality of our techniques separately from this crucial "Step 0".
> > >
> > > We now proceed to describe some of the main novelties in our proofs in the various settings under study - and we would of course be happy to include a version of this discussion in our revision as well.
> > >
> > > ---
> > >
> > > ### 1. Convergence rates under (RC), deterministic case
> > >
> > > A key step in the proof of Part 1 of Theorem 1 is to show that, if (MD) is run with the AdaMir step-size policy, the Bregman divergence $D(x^\ast,X_t)$ grows at most logarithmically in $t$ (a priori, this quantity is unbounded in domains with infinite Bregman diameter). In turn, this relies on the following series of arguments:
> > >
> > > - Showing that the Bregman residual $\delta_t$ is bounded (Lemma 1 in the paper). The only comparable result that we are aware of in the literature is Lemma 3.2 of [3] which, however, concerns a mirror-prox algorithm with two gradient queries per iteration, functions with bounded gradients, and a step-size defined in terms of a global norm. Related to our comment about the novelty of the definition of $\gamma_t$, the bound of Lemma 1 is only possible thanks to the exact form of $\gamma_t$: since the gradient of $f$ can become arbitrarily large in terms of global norms, bounding $\delta_t$ would not have been possible otherwise.
> > > - Using the definition of $\gamma_t$ to derive the bound $D(x^\ast,X_t) \leq D(x^\ast,X_1) + \sum_{s=1}^{t} [D(x_{s+1},X_s) + D(X_s,X_{s+1})] = D(x^\ast,X_1) + \sum_{s=1}^{t} \gamma_s^2 \delta_s^2$ (Lemma B.1 in the paper). The exact form of $\gamma_t$ also plays a vital role here, as it would not otherwise be possible to match the growth of $D(x^\ast,X_t)$ to $\sum_{s=1}^{t} \gamma_s^2 \delta_s^2$ in this precise way.
> > > - Using numerical sequence estimates (Lemma D.5 in the supplement) to bound $\sum_{t} \gamma_t^2 \delta_t^2$ as
> > >  $$\sum_{t=1}^{T} \gamma_t^2 \delta_t^2 = \sum_{t=1}^{T} \frac{\delta_t^2}{1 + \sum_{s=1}^{t-1} \delta_s^2} = \mathcal{O}(\log T)$$
> > >  This bound also hinges on the definition of $\gamma_t$ and the use of Lemma D.5 at _exactly this stage_ in the analysis (instead of using it to bound the weighted regret, as is typically the case in other approaches). We are not aware of a comparable technical result in the study of adaptive methods and, to the best of our knowledge, this represents a marked point of departure from the related literature.
> > >
> > > ---
> > >
> > > ### 2. Convergence rates under (RS), deterministic case
> > >
> > > When smoothness kicks in, the key ingredient is the summability of the residual sequence $\delta_t^2$ - or, equivalently, the stabilization of the adaptive step-size $\gamma_t$ to a strictly positive value.
> > >
> > > The first step here is relatively straightforward and relies on manipulating the descent inequality for relatively smooth functions [4] to write
> > > $$\frac{1}{2} \gamma_t \delta_t^2 \leq f(X_t) - f(X_{t+1}) + (L\gamma_t - 1/2) \gamma_t \delta_t^2$$
> > > and hence, after telescoping:
> > > $$\frac{1}{2} \sum_{t=1}^{T} \gamma_t \delta_t^2 \leq f(X_1) - \min f + \sum_{t=1}^{T} (L\gamma_t - 1/2) \gamma_t^2 \delta_t^2$$
> > >
> > > At this point, the approaches that we are aware of in the literature for Lipschitz smooth functions (in the ordinary sense) would branch out as follows to bound the RHS of the above estimate:
> > >
> > > - Use a gradient boundedness assumption - either _explicit_ [3] or _implicit_, stemming e.g., from the boundedness of the domain of interest [22,23]. However, this approach is problematic in the case of functions with gradient singularities and/or an unbounded domain because gradients typically grow unbounded.
> > > - Use prior knowledge of $L$ to enforce the condition $\gamma_t < 1/(2L)$ and show that $\sum_t \gamma_t \delta_t^2 < \infty$. [For example, this is the approach of [24] in unconstrained problems with $L$ known in advance (the case $\epsilon=0$ in the notation of said paper)] However, this approach is also problematic for our case, as we do not assume any prior knowledge of $L$.
> > >
> > > By contrast, we transcend both of these approaches by assuming _ad absurdum_ that $\gamma_t$ becomes vanishingly small as $t\to\infty$. By this stipulation, the RHS of the previous estimate is bounded above by some positive number; however, since the term $\sum_t \gamma_t \delta_t^2$ in the LHS is itself bounded from below by $1/\gamma_{T+1}$ (by the definition of the AdaMir step-size), the postulate $\gamma_t \to 0$ yields a contradiction and shows that $\delta_t^2$ is summable.
> > >
> > > We are not aware of any comparable approach in the literature.
> > >
> > >
> > > ---
> > >
> > > ### 3. Convergence rates in the stochastic case
> > >
> > > For the stochastic case, we use a different set of proof techniques altogether. The key challenge - and most novel part of our analysis - concerns the derivation of the explicit bound (14) that transitions smoothly from the $\sigma=0$ to the $\sigma>0$ regime.
> > >
> > > As in the deterministic case, the first step is relatively straightforward and relies on invoking the descent inequality for relatively smooth functions to write
> > >
> > > $$f(X_{t+1}) \leq f(X_t) + \langle g_t, X_{t+1} - X_t \rangle + \langle U_t, X_t - X_{t+1} \rangle + L [D(X_{t+1},X_t) + D(X_t,X_{t+1})]$$
> > >
> > > [As a note for the sequel, the residual term in the RHS of the above bound is crucially related to the definition of $\gamma_t$]
> > >
> > > The most challenging part of our analysis is how to deal with the "noise" term $\langle U_t, X_t - X_{t+1} \rangle$ since, due to measurability issues, we cannot simply hope to cancel it out by taking expectations. Here, our point of departure from the literature is to manipulate the specific definition of $\gamma_t$ to obtain $\langle U_t, X_t - X_{t+1} \rangle = \mathcal{O}(\sigma\gamma_t \delta_t)$ which, after summing from $1$ to $T$ yields the bound $\sum_{t=1}^T\langle U_t, X_t - X_{t+1} \rangle = \mathcal{O}(\sigma\sqrt{T})$. Thanks to this bound (which is uniform in $\omega$), we are subsequently able to lower bound the method's step-size as $\gamma_t \geq 1/(A+B\sigma\sqrt{t})$, where $A$ and $B$ are deterministic constants (cf. Lines 321-325 in our paper).
> > >
> > > Again, we are not aware of a similar approach in the literature.
> > >
> > >
> > > -----
> > >
> > > We hope that the above answers the reviewer's question on the originality of our proof techniques and their relation to existing works – thanks again for reaching out!

---

### Official Review · Reviewer_J6WU · 2021-07-15

**Rating:** 5
**Confidence:** 3

**Summary:**

- In the paper, the authors propose an adaptive parameter agnostic algorithm AdaMir for online and stochastic convex optimization problems.
- AdaMir is essentially like the popular Adagrad algorithm with Bregman distances involved in controlling the step-size.
- Their setting involves non-Lipschitz gradients, contrary to the traditional Lipschitz smooth setting.

**Limitations And Societal Impact:**

yes

**Main Review:**

Positives:
- The paper is well written, clear to read and the theory seems to be sound. I did not check the proofs though.
- Authors provide a good introduction to various concepts involved in the paper, for example, Bregman distances, relative smoothness and many others.
- AdaMir combines the best of two worlds, adaptive gradient methods and Mirror Descent based methods.
- In the deterministic setting, the ergodic convergence rates of AdaMir for both RC and RS settings are given in Theorem 1. Notably, optimal rates are obtained in both the settings.
- The stochastic setting is also considered, for which convergence rates are provided for AdaMir. Brief discussion on non-convex setting is also provided.


Negatives:
- The paper considers many settings which contributed to too much packed content in the paper. I wish authors focussed on just online setting itself, while relegating the stochastic setting to the supplementary material.
- No experiments were provided in the main paper. I would suggest moving the experiment from the supplement to the main paper.
- Example 1, 2, 3 can be moved to supplementary material. I think it is better to focus more on the AdaMir algorithm.
- The authors use a strongly convex $h$, which makes the definition of $\delta$ not so appealing.
- In my opinion, the main novelty lies in the definition of $\delta$ which makes use of two Bregman distance evaluations. However, I think the novelty in the rest of paper is not significant.
- More discussion on remark starting at line 118 is needed.
- A closely related work seems to be [45]. I think more discussion on this is needed preferably before just before the AdaMir algorithm. Also, discussion on regret bounds based on Proposition 3 is missing (please refer to [45]).
- What is the connection of your Definition of a Bregman function to the other standard definition using Legendre functions?
- There is no numerical evidence for the proposed method.

Other comments:
- In line 334, change "eay" to "easy".
- Typo in equation in line 229.


**Time Spent Reviewing:**

4

---

> ### Author Response · Authors · 2021-08-10
> **Replies to reviewer**
>
> We thank the reviewer for their time and remarks, and we are happy they appreciated the clarity of our presentation. We reply to the reviewer's main points below:
>
> 1. On the stochastic setting. We appreciate the reviewer's concern that the paper may appear "packed", but it is not possible to provide the proper context for rate interpolation between the deterministic and stochastic settings without keeping the stochastic analysis in the paper. In particular, we do not see how "focusing on the online case" would help, as this would not make the presentation (or the proofs) any simpler, whereas it would make the comparison to existing adaptive methods (like UnixGrad and AcceleGrad) significantly more difficult. We would be happy to include a remark about the online setting in our paper, but it is not otherwise possible to discuss questions pertaining to rate interpolation without also including the stochastic regime (especially since many of the envisioned applications of our work are stochastic in nature but not adversarial).
>
> 2. On moving the experiment sections to the main paper. Sure, we would be happy to use the extra page to do so, consider it done!
>
> 3. On moving Examples 1-3 to the appendix. We thought that these would be helpful to the reader, but we would be happy to provide just a pointer instead.
>
> 4. On the strong convexity of $h$. Strong convexity is not required (coercivity plus strict convexity work just as well for unbounded domains for example), but it simplifies the notation and presentation considerably.
>
> 5. On the novelty of the paper and the definition of $\delta$. While the introduction of the Bregman residual in the definition of $\delta$ is a major novelty of our paper, the analysis also includes a wide range of techniques that have never been used in the literature, and which are crucial to overcome domain boundedness and the lack of Lipschitz smoothness / continuity relative to a global norm. [For example, we are not aware of any work in the literature using the step-size stabilization techniques we employed in the proof of Proposition C.1.]
>
> 6. On Proposition 3 and the relevance of [45]. The confusion perhaps stems from L262: we should have said "starting point" not "key element". The inequality presented in Eq. 11 of Proposition 3 is a variant template inequality obtained for mirror descent methods after having divided both sides by $\gamma_{t}$. The main novelty is how to deal with the unboundedness of $D(x^{\ast},X_{t})$, which is where the particular definition of Adamir's adaptive step-size kicks in, and  ensures that $D(x^{\ast},X_{t})$ grows at most logarithmically in $T$. In [45], the authors study the "dual averaging" variant of mirror descent: the dual averaging formulation helps to bypass the issue of an infinite Bregman diameter in the *non-adaptive* case; however, it is otherwise very difficult to define an adaptive variant, precisely because dual averaging post-multiplies the gradients when they enter the algorithm. We will of course gladly comment on the above in our revision.
>
> 7. On the connection to Legendre functions. A Legendre function (with a compact zone/domain) has the defining property that $\|\nabla h(x) \| \rightarrow \infty$ whenever $x$ approaches the boundary of $\mathcal{X}$. In the context of our definition, this would correspond to the case $\mathrm{dom} \partial h = \mathrm{ri} \mathcal{X}$ where $\mathrm{ri} X$ denotes the relative interior of $\mathcal{X}$. We will be happy to include this remark in our revision.
>
> 8. On the lack of numerical evidence. As mentioned above, we would be happy to transfer the "numerical results" section of the supplement (Appendix E) to the main body of the paper. While on this issue, we would also like to point out that, in all our experiments, Adamir outperformed both state-of-the-art methods for the Fisher market example under study (entropic gradient descent and the proportional response algorithm).
>
> We hope and trust that the above addresses your remarks – and, of course, we will gladly incorporate all these remarks in our revision.

---

### Official Review · Reviewer_T495 · 2021-07-16

**Rating:** 7
**Confidence:** 4

**Summary:**

The paper proposed an adaptive mirror descent for convex optimization problems where Lipschitz continuity or smoothness may be not available, both deterministic and stochastic settings are discussed.


**Limitations And Societal Impact:**

See above for limitations and no negative societal impact.


**Main Review:**

The paper provides a detailed literature review on the considered problem. The study is well motivated and the results obtained are clearly stated and discussed.

Please be clear about the $O(1/T^\alpha) $ convergence rates appeared in the paper. For Nesterov’s accelerated gradient, the $O(1/T^2)$ rate is on the objective function value at the last iterate, while for some other methods mentioned in the paper, $O(1/T^2)$ rate is for ergodic averaged point.





**Time Spent Reviewing:**

6

---

> ### Author Response · Authors · 2021-08-10
> **Replies to reviewer**
>
> Thank you for your encouraging remarks and your positive evaluation! We reply to your precise comments below:
>
> - On the $O(1/T^2)$ rate for accelerated methods. Indeed, in UnixGrad for example, there is an averaging process involved in generating the points where the function's gradients are queried. However, even in this case, the rate provided concerns the last point where the function's gradient was queried – i.e., again, the method's last iterate. In other methods, this is perhaps less clear, but we will be happy to include more details for each of the accelerated methods discussed.
>
> We hope that the above answers your question, and we will be sure to include this remark in our revision. Thank you again for your thoughtful review and encouraging assessment!

---

### Official Review · Reviewer_jyep · 2021-07-17

**Rating:** 6
**Confidence:** 4

**Summary:**

The paper proposes an adaptive mirror descent method for solving relative smooth optimization problems. The idea of using adaptive stepsize for first-order methods is not new in the literature, but the proposed adaptive stepsize in this paper is new.

**Limitations And Societal Impact:**

N.A.

**Main Review:**

The paper is well-written and the obtained convergence rate in this paper is nice. My main concern is the well-definedness of the proposed algorithm. Proposition 2 proves that $x^+ \in dom  \partial h$ for all $x\in dom  \partial h$ and $g\in \mathcal V^*$. But it is not clear if $x^+$ belongs to $dom \partial f$ or not. This is crucial to guarantee the existence of the next $g$.

Some minor comments:

1. Provide the strongly convex constant $K$ for the examples in lines 157-165.
2. Typo in (RC) (see line 171): $x\in dom \partial h, x' \in dom h$.
3. Definition 3: it should be "relative smooth to $h$".
4. Proposition 1: provide where $x$ and $x'$ belong in each inequality.
5. Section 4.1: is it "prox-mapping" or "prox-gradient/subgradient-mapping"?


**Time Spent Reviewing:**

18

---

> ### Author Response · Authors · 2021-08-10
> **Replies to reviewer**
>
> Thank you for the detailed review, and for your positive and encouraging remarks! We reply to your precise questions below:
>
> 1. On the algorithm's well-posedness. Apologies for the confusion, $\mathcal{X}$ in Definition 1 should instead read $\mathcal{X}_{\circ}$, which implies in particular that $\mathrm{dom} \partial h \subseteq \mathrm{dom} \partial f$. Thanks a lot for spotting this, we will of course correct the typo.
>
> 2. Minor comments 1-4. We will take care of these, thanks for spotting them.
>
> 3. On the use of the term "prox-mapping" (minor comment 5).  Our terminology follows Nemirovski et al. (SIOPT, 2009; ref. 30 in our paper). To the best of our knowledge, there is no consensus in the literature on how to call this mapping, but "prox-mapping" seems to be the most commonly used term. We did not use terms like "proximal gradient" or "proximal subgradient" to avoid any confusion with proximal point methods for composite optimization problems.
>
> We hope that the above answers your questions, and we will be sure to include them in our revision. Thank you again for your thoughtful review and positive evaluation!

---

### Decision · Program_Chairs · 2021-09-27

**Decision:**

Accept (Poster)

**Comment:**

Overall the reviewers liked the paper and found the contributions strong enough for NeurIPS. I have read the paper, and while I have an issue with the claim of the authors that their algorithm is adaptive, while it requires the specification of the "smoothing function" h(), I still appreciate the overall technical contribution provided in the analysis and obtained results.